# TLR7 activation at epithelial barriers promotes emergency myelopoiesis and lung antiviral immunity

**William D Jackson[1†], Chiara Giacomassi[1†‡], Sophie Ward[1], Amber Owen[2], Tiago C Luis[1], Sarah Spear[3], Kevin J Woollard[1], Cecilia Johansson[2], Jessica Strid[1], Marina Botto[1]\***

[1]Centre for Inflammatory Disease, Department of Immunology and Inflammation, Imperial College London, London, United Kingdom; [2]National Heart and Lung Institute, Imperial College London, London, United Kingdom; [3]Division of Cancer, Department of Surgery and Cancer, Imperial College London, London, United Kingdom

**\*For correspondence:**
m.botto@imperial.ac.uk

[†]These authors contributed equally to this work

**Present address:** [‡]William Harvey Research Institute, Queen Mary University of London, London, United Kingdom

**Competing interest:** The authors declare that no competing interests exist.

**Abstract** Monocytes are heterogeneous innate effector leukocytes generated in the bone marrow and released into circulation in a CCR2-dependent manner. During infection or inflammation, myelopoiesis is modulated to rapidly meet the demand for more effector cells. Danger signals from peripheral tissues can influence this process. Herein we demonstrate that repetitive TLR7 stimulation via the epithelial barriers drove a potent emergency bone marrow monocyte response in mice. This process was unique to TLR7 activation and occurred independently of the canonical CCR2 and CX3CR1 axes or prototypical cytokines. The monocytes egressing the bone marrow had an immature Ly6C-high profile and differentiated into vascular Ly6C-low monocytes and tissue macrophages in multiple organs. They displayed a blunted cytokine response to further TLR7 stimulation and reduced lung viral load after RSV and influenza virus infection. These data provide insights into the emergency myelopoiesis likely to occur in response to the encounter of single-stranded RNA viruses at barrier sites.

## Editor's evaluation

This work advances our understanding of TLR7 signalling at epithelial surfaces that drives monocytes expansion and its impact on viral responses. The evidence supporting this conclusion is solid, particularly data demonstrating TLR7 stimulation and the requirement for TLR7 in the monocyte expansion. The work will be of interest to immunologists and virologists.

## Introduction

Monocytes are circulating, short-lived mononuclear phagocytes critical for host defence against infection. In mice, there are at least two subpopulations of blood monocytes defined by their expression of lymphocyte antigen 6C (Ly6C) (*Geissmann et al., 2003*). Ly6C-high monocytes are 'classical' inflammatory monocytes, which express high levels of CCR2, low levels of CX3CR1, and respond to canonical bacterial cues such as lipopolysaccharide (LPS). Ly6C-low monocytes are defined as 'non-classical', expressing low levels of CCR2 and high levels of CX3CR1 (*Geissmann et al., 2003*). These non-classical monocytes patrol the vascular lumen during times of homeostasis, surveying its integrity and orchestrating the disposal of damaged endothelial cells and subsequent inflammatory response (*Auffray et al., 2007*; *Carlin et al., 2013*; *Turner-Stokes et al., 2020*). This patrolling behaviour is

independent of the normal leukocyte adhesion cascade and requires firm adhesion via the β2-integrin LFA1 (*Carlin et al., 2013*). Non-classical monocytes can respond directly to viral cues via toll-like receptor 7 (TLR7), yet respond poorly to LPS (*Cros et al., 2010*). The existence of a heterogeneous, MHC-II-high-intermediate population has also been suggested (*Menezes et al., 2016*; *Mildner et al., 2017*), but in mice the functional distinction of intermediate monocytes remains unclear. Despite widespread usage, this classification system may oversimplify monocyte heterogeneity as mass cytometry has identified up to eight subpopulations of monocytes in healthy human blood (*Hamers et al., 2019*) and novel subpopulations can emerge during inflammation and fibrosis (*Satoh et al., 2017*), such as Sca-1-positive 'emergency monocytes' during parasite infection (*Abidin et al., 2017*; *Askenase et al., 2015*).

In steady-state conditions, monocytes are produced in the bone marrow (BM) where they are derived from commitment of haematopoietic stem and progenitor cells (HSPCs) along a defined pathway that culminates in terminally differentiated mature monocytes. Although still debated, the most widely accepted of these pathways is the sequential commitment of HSPCs to the common myeloid progenitor (CMP), the monocyte-dendritic cell progenitor (MDP), and the common monocyte progenitor (cMoP), before finally generating BM monocytes (*Wolf et al., 2019*) in a process that is dependent on the colony-stimulating factor 1 (CSF1) and the transcription factor PU.1 (*DeKoter et al., 1998*). In addition, the work of Goodridge and colleagues suggests a functionally distinct monocyte population that can be derived from the granulocyte-monocyte progenitor (GMP), bypassing the MDP and the cMoP (*Yáñez et al., 2017*). The first monocyte population to be produced in the BM are the Ly6C-high monocytes, which during homeostasis are obligate precursors for Ly6C-low monocytes in a CEBP-β dependent process (*Mildner et al., 2017*). This is consistent with previous findings using transgenic fate mapping mice (*Yona et al., 2013*), direct adoptive transfer of Ly6C+ monocytes (*Varol et al., 2007*; *Yona et al., 2013*), and re-population kinetic studies following depletion regimes (*Sunderkötter et al., 2004*). The same process has also been suggested to occur in humans (*Patel et al., 2017*). Ly6C-high monocytes have a circulating half-life of <1 d in both mice and humans before converting through an intermediate stage into Ly6C-low monocytes, which can remain in the blood for between ~2 and 7 d (*Patel et al., 2017*; *Yona et al., 2013*). The egression of Ly6C-high monocytes from the BM is largely dependent on the C-C chemokine receptor type 2 (CCR2) (*Tsou et al., 2007*). Similarly, Ly6C-high monocyte extravasation into tissues is mediated by CCR2 in response to local production of the C–C motif chemokine ligand 2 (CCL2) or 7 (CCL7) (*Shi and Pamer, 2011*). However, the dynamics of monocyte subpopulation production and BM egression during inflammation and/or infection remain poorly understood.

During infection or tissue injury, monocytes are crucial for controlling the invading pathogen (*Haist et al., 2017*; *Shi and Pamer, 2011*) or for regulating the tissue repair process (*Wynn and Vannella, 2016*). For example, the recruitment of Ly6C-high inflammatory monocytes in the lungs in response to respiratory pathogens like respiratory syncytial virus (RSV) is essential to control viral load and lessen disease severity (*Goritzka et al., 2015*). As such, the process of myelopoiesis is differentially regulated during infection or inflammation to rapidly meet demand for 'emergency' effector monocytes or neutrophils. An acute requirement for additional monocytes can either be met by the spleen via a reservoir of mature splenic monocytes and accompanying extramedullary haematopoiesis (*Swirski et al., 2009*) or by conventional BM haematopoiesis (*Wolf et al., 2019*). A key component of the 'emergency' process is sensing and communicating the danger signals to the haemopoietic progenitor pool in either the BM or the spleen. This occurs primarily through activation of pattern recognition receptors (PRRs), most notably the TLR family, whose expression has been confirmed on HSPCs in both human and mouse (*Nagai et al., 2006*; *Sioud et al., 2006*). The sensing of pathogens by the haematopoietic system can either occur directly or as a secondary effect of inflammatory mediators produced at the barrier sites such as the skin or the gut (*Askenase et al., 2015*; *Baldridge et al., 2010*; *Pietras et al., 2016*) or by BM stromal cells (*Shi and Pamer, 2011*). The nature of the distal signals from barrier sites to BM and the features of the 'emergency' processes triggered under different settings remain largely unknown.

The skin is the largest barrier site in the body and is targeted as an entry point by a variety of pathogens, perhaps most notably the diverse group of arboviruses spread by mosquitoes. These infect hundreds of millions of people annually and include serious threats to human health such as dengue virus and Zika virus, both of which are single-stranded RNA viruses sensed by TLR7 (*Paixão

*et al., 2018*). Recently, it has been shown that the skin immune response via TLR7 expressed in dermal dendritic cells can locally protect against a second viral infection at the inoculation site (*Bryden et al., 2020*). However, it is unclear what effect TLR7 activation at the epithelial barrier has on the haematopoietic system and the subsequent innate immune response in other organs. Here we used R848, a TLR7/8 agonist, and demonstrate that only persistent TLR7 stimulation at an epithelial barrier such as the skin or gut was able to drive a unique CCR2-independent BM emergency monocyte response. This process was characterised by the release of immature Ly6C-high pre-monocytes into the periphery and their differentiation to both Ly6C-low monocytes in the blood and to tissue macrophages in multiple organs. The emergency monocytes released by the BM under these conditions displayed an impaired response to TLR7 restimulation and promoted lung viral control, which dampened disease severity after RSV and influenza virus infection.

## Results

### Persistent TLR7 stimulation at epithelial barrier sites drives systemic monocytosis

To investigate how the haematopoietic response to a single-stranded RNA virus could be influenced by the point of entry, we administered a TLR7 agonist, R848, to BALB/c mice topically or intraperitoneally (IP) (100 µg, three times a week for 4 wk, equivalent to 12 treatments) (*Figure 1—figure supplement 1*) and observed a marked monocytosis only in the topical R848 group (*Figure 1A*). While blood monocyte counts (CD11b+CD115+) in vehicle and IP-treated mice remained largely unchanged (mean of 850.8 ± 159/µl), mice that received topical R848 had a mean monocyte count of 25,154 ± 2956/µl, an approximately 30-fold expansion (*Figure 1A*). The monocytosis was accompanied by a ratio switch in the Ly6C-high and Ly6C-low monocyte subpopulations, with the Ly6C-low population accounting for >90% of total monocytes after topical R848 (*Figure 1A*). We also found a pronounced splenomegaly in the topically treated mice which was not present in the mice receiving R848 IP (*Figure 1—figure supplement 1*). To investigate the kinetics of the monocytosis, mice were treated with topical R848 four times and blood samples were taken 24 hr after each treatment. After one treatment, we observed a marked drop in monocyte (*Figure 1B*) and lymphocyte (both CD3+ T cell and B220+ B cell) counts (*Figure 1—figure supplement 1*). While the lymphocyte counts slowly returned within the normal range by four treatments (*Figure 1—figure supplement 1*), both monocyte subpopulations expanded quickly after two treatments (*Figure 1B*). On subsequent R848 applications, the Ly6C-low monocyte population continued to expand, whilst the level of Ly6C-high monocytes remained stable (*Figure 1B*), ultimately leading to a ratio switch (*Figure 1C*) as in *Figure 1A*. Throughout the time course, no change was seen in blood neutrophil counts (CD11b+Ly6G+Ly6C-low), indicating that topical R848 does not trigger a pan-myeloid response (*Figure 1D*). To confirm that the monocyte response was not due to the systemic diffusion of the compound, we administered R848 intravenously (IV) and compared it to topical application. While both routes of administration caused lymphopenia, only the topical route was able to promote monocytosis and a ratio switch toward Ly6C-low monocytes (*Figure 1E*). We then reasoned that perhaps the monocyte response triggered by TLR7 activation was specific to epithelial barrier sites. To test if this was a unique response to topical challenge, R848 was dissolved in the drinking water at a concentration that was calculated to be equivalent to the dose applied topically (*Bachmanov et al., 2002*). Oral R848 induced a monocytosis with a proportional switch toward Ly6C-low monocytes that was comparable to that seen after topical treatment (*Figure 1F*).

As the response to other TLR stimuli such as LPS is dose-dependent and repeated administrations can either result in sensitisation or tolerance, which is considered a form of 'innate immune memory' (*Biswas and Lopez-Collazo, 2009*), we performed a dose–response experiment with R848 given IP. None of the doses tested caused a significant increase in blood monocyte counts when compared to vehicle (*Figure 1—figure supplement 1*). In addition, the monocytosis following topical R848 was clearly dose dependent, with 100 µg inducing a strong response, 1 µg causing no change, and the 10 µg group displaying an intermediate phenotype (*Figure 1—figure supplement 1*). Consistent with the dose–response data, application of imiquimod (IMQ), which is an ~100-fold less potent agonist, did not elevate peripheral monocyte counts even after six daily treatments (*Figure 1—figure supplement 1*). Together these data demonstrate that a persistent and potent TLR7 stimulation at

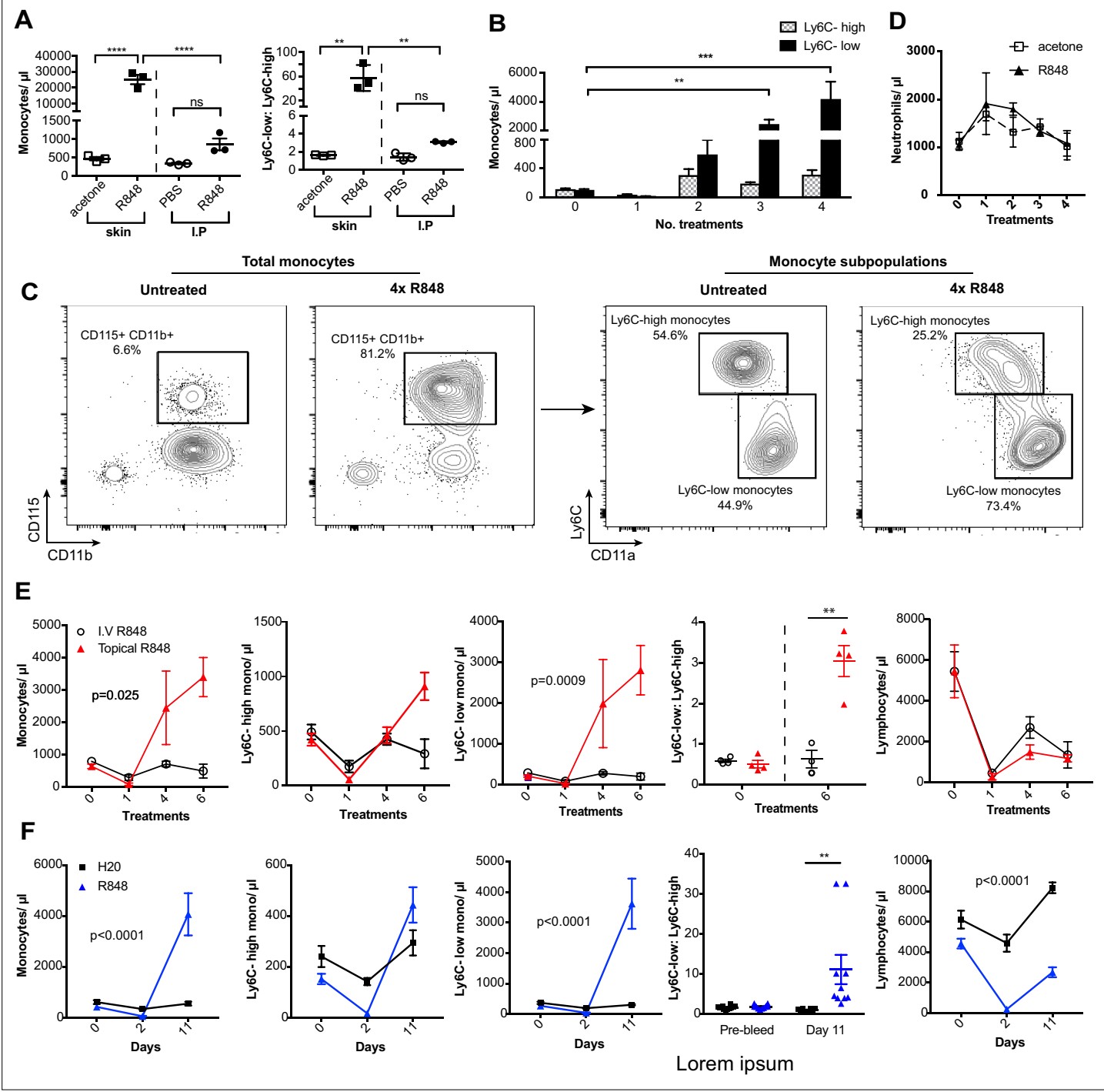

**Figure 1.** Repetitive R848 administration at barrier sites drives a profound monocytosis. (**A**) BALB/c mice (n = 3 per group) received 100 µg of R848 topically or intraperitoneally (IP), 3× per week for 4 wk. Control BALB/c mice were given topical acetone or 200 µl of PBS IP. Left panel shows the number of total monocytes (CD11b+CD115+), the right panel the ratio between Ly6C-high and Ly6C-low monocytes. (**B**) BALB/c mice (n = 3 per group) received four treatments of topical R848 (100 µg). Numbers of Ly6C-high (grey bars) and Ly6C-low (black bars) monocyte 24 hr after each treatment. (**C**) Representative plots of total monocytes (left panel) and of subpopulations (right panel) in mice treated topically with acetone or R848 four times. (**D**) Mice treated as in (**B**). Neutrophil counts 24 hr after each treatment. (**E**) BALB/c mice (n = 4 per group) received six treatments with 100 µg topical (red line) or IV R848 (black line). Blood counts are shown for total monocytes, Ly6C-high monocytes, Ly6C-low monocytes, monocyte subpopulation ratio and lymphocytes at 24 hr after the indicated treatment. (**F**) C57BL/6 mice were given drinking water containing 8.3 µg/ml R848 (blue line, n = 10) or vehicle control (black line, n = 12) for 11 d. Blood counts are shown for total monocytes, Ly6C-high monocytes, Ly6C-low monocytes, monocyte subpopulation ratio and lymphocytes at the indicated time point. Data representative of at least two independent experiments (except **A**). One-way ANOVA with

*Figure 1 continued on next page*

*Figure 1 continued*

Bonferroni's multiple-comparison test (**A**); two-way ANOVA with Tukey's multiple comparison for analysis of time-course experiments (**B, D–F**). Data are the mean ± SEM; only significant p-values are indicated; **p<0.01; ***p<0.001; ****p<0.0001.

The online version of this article includes the following source data and figure supplement(s) for figure 1:

**Source data 1.** FACS raw data.

**Figure supplement 1.** The effects of TLR7 stimulation depend on dose and route of administration.

an epithelial barrier can trigger a distinctive innate immune response characterised by a profound monocytosis.

## Skin-induced monocytosis requires TLR7 activation

We next sought to investigate whether the monocyte response via an epithelial barrier could also be promoted by other TLR stimuli. We therefore treated mice topically with LPS or poly I:C (TLR4 and TLR3/RIG-I agonists, respectively), alongside an R848 control group. For this experiment, we utilised a modified water-soluble version of R848 to allow all agonists to be dissolved in the same vehicle. As previously, topical R848 triggered an immediate leukopenia followed by the characteristic monocytosis dominated by Ly6C-low monocytes, whereas LPS and poly I:C had no substantial effects on blood cell counts (*Figure 2A*). Similarly, topical application of a TLR9 agonist, CpG oligodeoxynucleotides (ODNs), neither changed monocyte counts nor caused lymphopenia (*Figure 2B*). Moreover, topical application of the potent pro-inflammatory stimulus 12-O-tetradecanoylphorbol-13-acetate (TPA) did not affect monocyte counts after four treatments (*Figure 2C*). In view of the marked difference between the effects induced by R848 and the other TLR agonists, we next used TLR7-deficient ($Tlr7^{-/-}$) mice to exclude activation of TLR8 or inflammasome, or any off-target effects of the R848 compound. Total monocytes and Ly6C-low monocytes were significantly elevated in the wild-type (WT) animals, while $Tlr7^{-/-}$ mice did not show a response to R848 confirming the specificity of the pathway involved (*Figure 2D*).

Various cytokines have previously been implicated in the regulation of myeloid cell production during inflammation, most notably the interferon (IFN) family and IL1β (*Askenase et al., 2015*; *Buechler et al., 2013*; *Mitroulis et al., 2018*). We found no obvious differences in monocyte counts after topical R848 treatments in either IFN-y- or IFNAR1-deficient mice compared to the WT animals (*Figure 2—figure supplement 1*), discounting a role for type I and type II IFNs. We then utilised the IL-1 receptor antagonist anakinra, which has been reported to cross-react with the mouse protein (*Iannitti et al., 2016*), and observed no differences in the induction of monocytosis or the monocyte subpopulation ratio (*Figure 2—figure supplement 1*), making the involvement of IL-1 unlikely. In addition, blocking IL-6 and TNF-α signalling did not prevent the onset of monocytosis (*Figure 2—figure supplement 1*). These data collectively suggest that only activation of the TLR7 pathway at the skin barrier can drive the changes in the myeloid compartment that occur independently from inflammasome activation or some of the well-established cytokine-mediated pathways.

## Myeloid cells orchestrate the R848-induced monocytosis

We next attempted to determine which cells initiate the monocyte response. We first generated BM chimeras using $Tlr7^{-/-}$ mice (*Hemmi et al., 2002*) to distinguish between stromal and haematopoietic cells. BM reconstituted mice were treated with topical R848 and monocyte response assessed. After four treatments, there was a marked increase in the proportion of total monocytes and Ly6C-low monocytes in the mice that received WT BM, but not in mice that received $Tlr7^{-/-}$ BM, regardless of the host genotype (*Figure 3—figure supplement 1*), indicating that cells of haematopoietic origin and not irradiation-resistant skin-resident cells like stromal or epithelial cells were responsible for the myeloid response to R848.

To identify the BM-derived cells responding to the TLR7 activation, we first utilised $Rag2^{-/-}$ mice, which lack both T- and B cells (*Shinkai et al., 1992*). After four topical treatments with R848, both WT and $Rag2^{-/-}$ animals developed a monocytosis dominated by Ly6C-low cells (*Figure 3A*), indicating that lymphocytes are dispensable. We then crossed $Tlr7$-floxed mice (*Solmaz et al., 2019*) with $Lyz2^{Cre/Cre}$ animals, which express the Cre recombinase in monocytes, neutrophils, and some macrophage populations (*Abram et al., 2014*). While after R848 treatment both WT and $Lyz2^{Cre/+} × Tlr7^{fl}$ mice developed

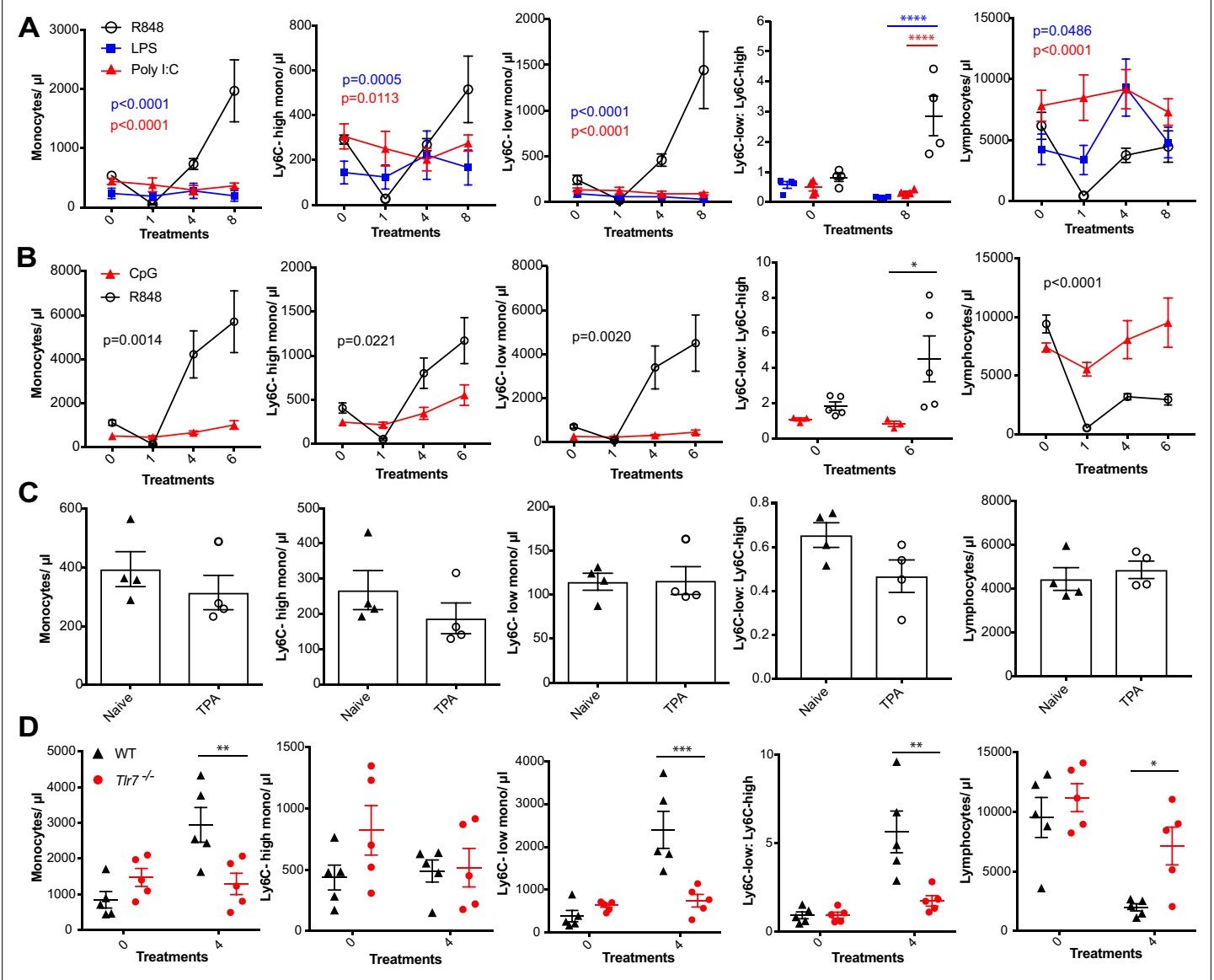

**Figure 2.** R848-induced monocytosis is specific to TLR7 activation. Blood counts are shown for total monocytes, Ly6C-high monocytes, Ly6C-low monocytes, monocyte subpopulation ratio, and lymphocytes 24 hr after the indicated treatment. (**A**) BALB/c mice (n = 4 per group) received eight treatments with topical R848 (100 μg, black line), LPS (100 μg, blue line), or Poly l:C (100 μg, red line). (**B**) C57BL/6 mice received six treatments with topical R848 (100 μg, n = 5, black line) or CpG (100 μg, n = 3, red line). (**C**) BALB/c mice (n = 4 per group) received four topical treatments with 2.5 nmol TPA. (**D**) C57BL/6 mice (n = 5, black triangles) or *Tlr7*⁻/⁻ mice (n = 5, red circles) received four treatments with topical R848. Data representative of two independent experiments (except **A** and **C**). Two-way ANOVA with Tukey's multiple-comparison for time-course experiments (**A, B**); unpaired *t*-test (**C, D**). Data are the mean ± SEM; only significant p-values are indicated; *p<0.05; **p<0.01; ***p<0.001.

The online version of this article includes the following source data and figure supplement(s) for figure 2:

**Source data 1.** FACS raw data.

**Figure supplement 1.** IFNs and cytokines are not involved in the R848-induced monocytosis.

monocytosis, this expansion was reduced by >50% in the mice lacking TLR7 in *Lyz2*-expressing cells (*Figure 3B*), demonstrating that Lyz2-expressing cells were at least partially responsible for the skin response to TLR7 activation. As mast cells, eosinophils, and basophils are not reported to express *Lyz2* (*Abram et al., 2014*), we investigated these cells individually. *ΔdblGATA* mice, lacking eosinophils, and *Cpa3*^Cre/+ mice, lacking both mast cells and basophils, did not show any obvious defect in their response to R848 (*Figure 3—figure supplement 1*). In addition, neutrophils depletion using an anti-Ly6G depletion antibody (*Daley et al., 2008*) failed to abolish the monocytosis. On the contrary, it

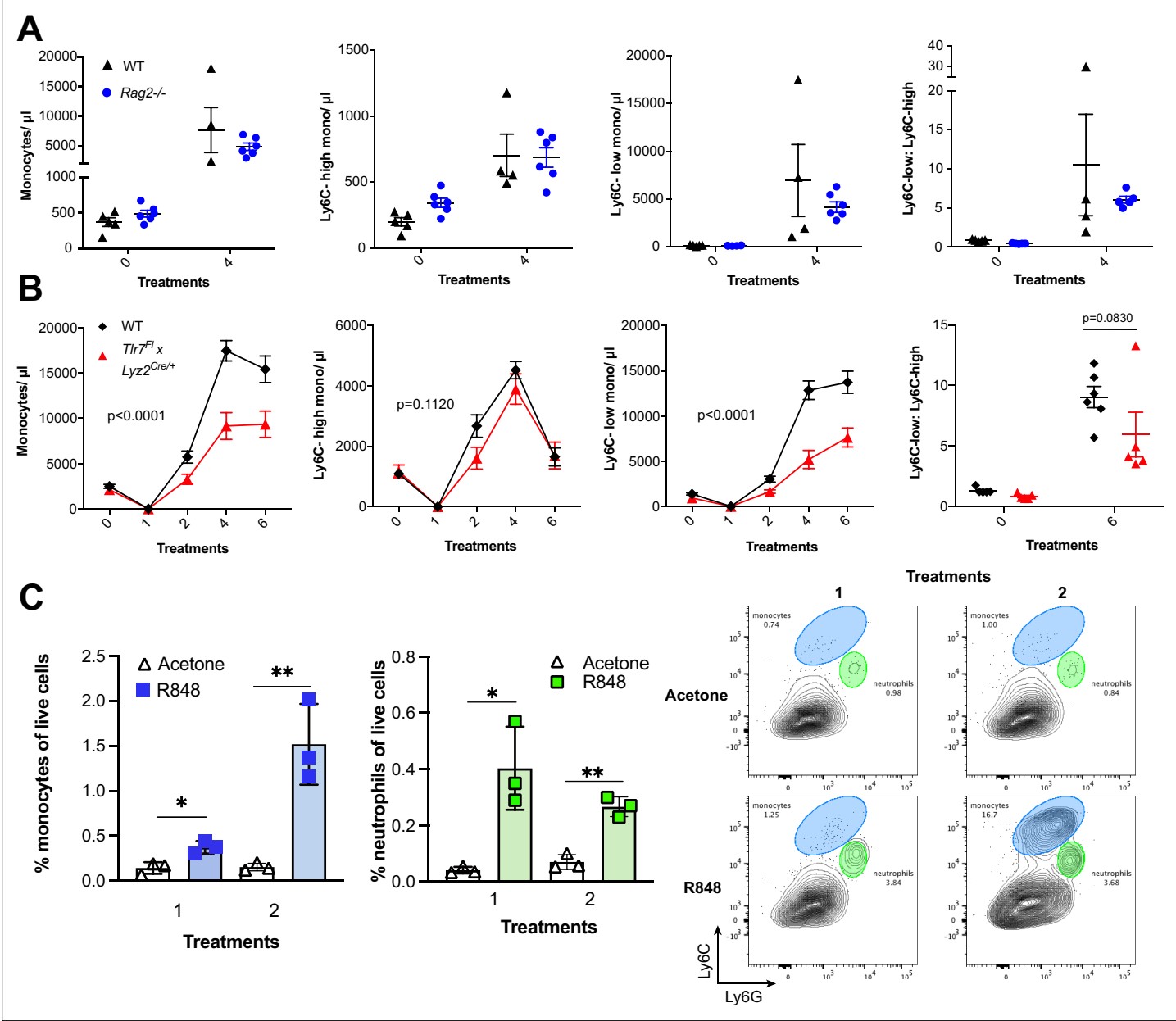

**Figure 3.** R848-induced monocytosis is driven by TLR7 activation of myeloid cells. (**A, B**) Blood counts for total monocytes, Ly6C-high monocytes, Ly6C-low monocytes, monocyte subpopulation ratio, and lymphocytes at baseline and 24 hr after the indicated treatment. (**A**) C57BL/6 mice (n = 5, black triangles) and *Rag2*-/- mice (n = 6, blue circles) received four topical treatments with R848. (**B**) C57BL/6 mice (n = 6, black rhombi) and *Tlr7<sup>fl</sup> × Lyz2<sup>Cre/+</sup>* mice (n = 5, red triangles) received six topical treatments with R848. (**C**) C57BL/6 mice (n = 3 per group) were treated once or twice with topical R848 or acetone. Treated ear skin was harvested at 24 hr post-treatment and anal- ysed by flow cytometry. Proportion of monocytes (CD11b+Ly6C+Ly6G-low) and neutrophils (CD11b+Ly6G-high- Ly6C-low) among total live cells (left panels). Representative flow cytometry plots gated on CD11b+ cells (right panels). Data represent a single experiment (**A, C**) or two experiments (**B**). Two-way ANOVA, with Tukey's multiple-comparison for time-course experiments (**A, B**); unpaired *t*-test (**C**). Data are the mean ± SEM; only significant p-values are indicated; *p<0.05; **p<0.01.

The online version of this article includes the following source data and figure supplement(s) for figure 3:

**Source data 1.** FACS raw data.

**Figure supplement 1.** The role of myeloid cell populations in R848-driven monocytosis.

increased the R848-induced monocytosis (*Figure 3—figure supplement 1*) consistent with the observation that (*Cortez-Retamozo et al., 2012*; *Swirski et al., 2009*) neutrophil depletion can trigger a mild degree of monocytosis (*Patel et al., 2019*). In light of these observations, we concluded that a BM-derived myeloid population was contributing to promote the systemic response following TLR7 activation.

We next explored whether local tissue-infiltrating myeloid cells were involved in the systemic response. We applied R848 twice and observed infiltration of monocytes in the skin after the first treatment (*Figure 3C*), suggesting that these BM-derived myeloid cells were not only involved in triggering the initial response to TLR7 stimulation, but also acted in a positive feedback manner and played a key role in maintaining the systemic myelopoiesis.

## The emergency monocyte response triggered by cutaneous R848 originates in the BM

While haematopoiesis of myeloid cells during homeostasis occurs predominantly in the BM, under pathological conditions this can be superseded by extramedullary haematopoiesis in the spleen (*Cortez-Retamozo et al., 2012*; *Swirski et al., 2009*). To understand the source of the monocyte response triggered by cutaneous TLR7 activation, we performed splenectomy or sham surgery and applied topical R848 treatment after the mice had fully recovered from the surgery. Monocytosis and a ratio switch towards Ly6C-low monocytes occurred in the R848-treated mice with or without spleen and were markedly different from the untreated splenectomised mice (*Figure 4A*), indicating that the stimulation of TLR7 at the skin barrier was driving a haematopoietic response mainly in the BM. Of note, the R848-treated splenectomised mice showed a slightly enhanced phenotype, suggesting that the spleen may retain some of the circulating monocytes. To confirm the BM origin of the R848-induced monocytosis, we conducted in vivo fate-tracing experiments using HSC-SCL-CreER$^{T}$;R26EYFP mice (*Göthert et al., 2005*). Under our experimental conditions, tamoxifen administration induced recombination in ~45% of the BM HSCs (*Figure 4—figure supplement 1*) with no recombination in the peripheral blood. Consistent with previous experiments, the tamoxifen-treated HSC-SCL-CreER$^{T}$;R26EYFP mice responded to the topical R848 with a marked increase in the total monocyte count and a ratio switch towards Ly6C-low monocytes (*Figure 4B*), indicating that neither the genetic modification nor the tamoxifen had altered the BM response. Importantly, the same pattern was observed among the EYFP-positive monocytes that were predominantly Ly6c-low monocytes (*Figure 4C*), confirming that the R848-induced monocytes originates mainly from the BM.

Monocytes are derived from a defined program of progenitor differentiation in the BM outlined in *Figure 4D*. To investigate this pathway, we assessed proliferation of BM progenitor cells using bromodeoxyuridine (BRDU) pulse-chase experiments. When we examined BRDU incorporation in progenitor cells (gated strategy in *Figure 4—figure supplement 1*), we found that R848 dramatically increased the percentage of HSPCs in cell cycle from ~4.6% to ~45.6% (*Figure 4E and F*). Consistent with the peripheral blood data showing that TLR7 activation targets the monocytes and does not affect the granulocyte lineage, there was a significant increase in BRDU positivity among MDPs and cMoPs, but not in GMPs (*Figure 4E*). Unsurprisingly, the mature BM monocyte population also showed increased BRDU incorporation (*Figure 4E*) and this was present in both monocyte subpopulations indicating that the effects of TLR7 activation were not limited to the Ly6C-low cells (*Figure 4E*). Of note, the R848-induced bone marrow monocytes also upregulated expression of the stem cell antigen 1 (Sca1) (*Figure 4G*), a feature consistent with an emergency myelopoiesis (*Askenase et al., 2015*).

Given the dramatic increase in the proportion of normally quiescent HSPCs in cell cycle, we hypothesised that R848-induced monocytosis would persist for a considerable period after cessation of the treatment. Indeed, after four topical R848 treatments, it took at least 17 d for total monocyte counts to return to pre-bleed levels (*Figure 4H*). The monocyte changes were accompanied by a decrease in blood lymphocyte numbers, which reached pre-treatment levels by day 10 (*Figure 4H*). We therefore concluded that cutaneous TLR7 activation can bypass the splenic reservoir and can instruct the BM HSPCs to proliferate and differentiate predominantly along the monocytic pathway triggering an emergency myelopoiesis.

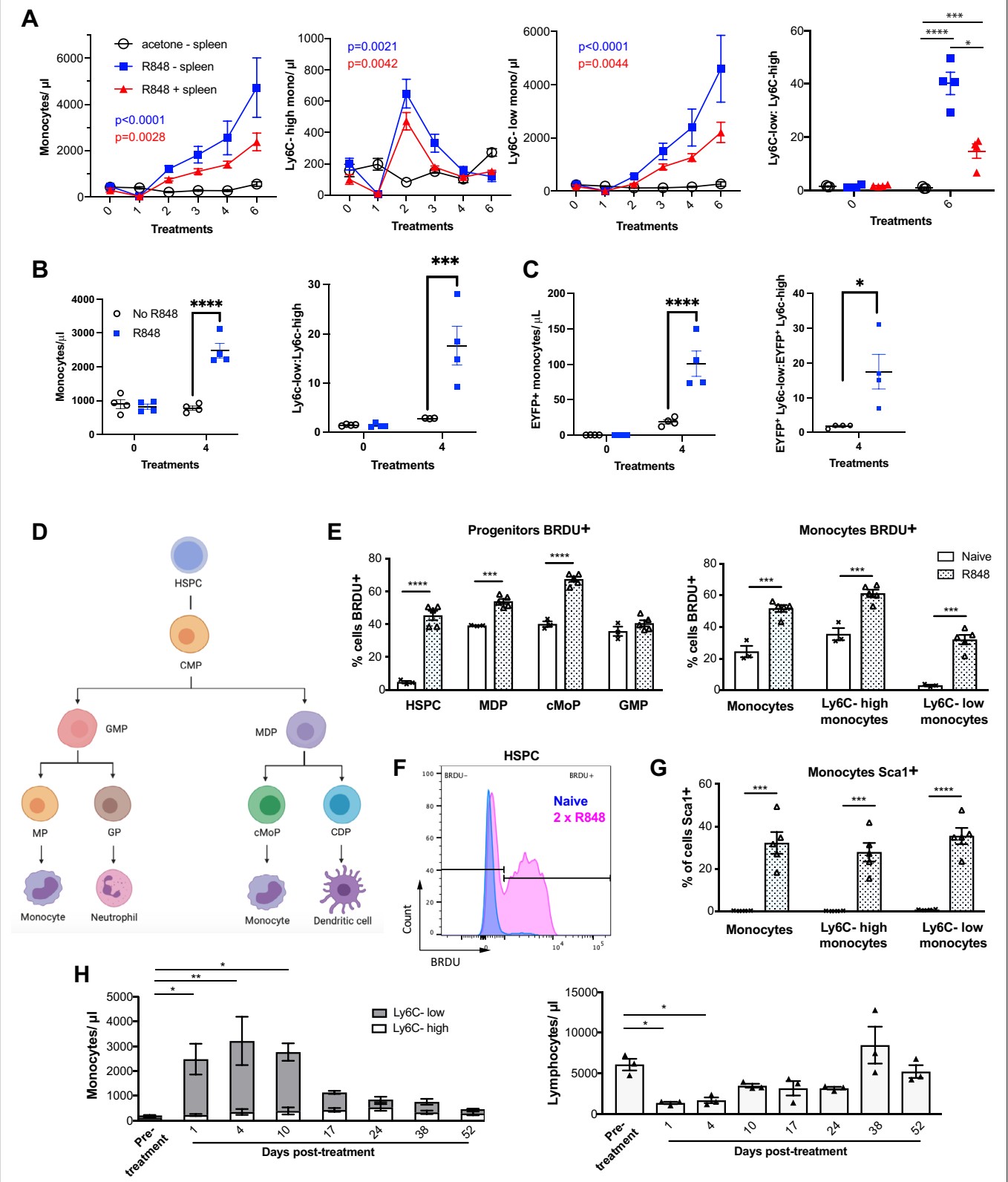

**Figure 4.** R848-induced monocytes are derived from the bone marrow (BM) and have features of emergency myelopoiesis. (**A**) BALB/c mice underwent splenectomy or sham surgery and were left to recover for 7 wk. Among the splenectomised mice, a group was treated 6× with topical R848 (n = 4, blue line) and the remaining mice with acetone (n = 3, solid black line). The sham surgery group was treated with R848 (n = 4, red line). Blood counts are shown for total monocytes, Ly6C-high monocytes, Ly6C-low monocytes, and the monocyte subpopulation ratio at baseline and 24 hr after the

*Figure 4 continued on next page*

*Figure 4 continued*

last treatments. (**B, C**) HSC-SCL-Cre-ERT;R26R-EYFP mice (n = 4 per group) received tamoxifen (4 mg/100 μl) by oral gavage for five consecutive days. Three days later, the left ears were treated topically 4× with R848 (blue squares) or left untreated (open circles). Shown are the total blood monocyte counts and the monocyte subpopulation ratio (**B**); the total blood EYFP+ monocyte counts and the subpopulation ratio among the EYFP-expressing monocytes at baseline and 24 hr after the last treatment (**C**). (**D**) Diagram illustrating BM myeloid progenitor differentiation: haematopoietic stem and progenitor cells (HSPC); common myeloid progenitor (CMP); monocyte-dendritic precursor (MDP); granulocyte-monocyte progenitor (GMP); common monocyte progenitor (cMoP); common dendritic progenitor (CDP); monocyte-committed progenitor (MP); granulocyte-committed progenitor (GP). (**E–G**) C57BL/6 mice were injected with 2 mg BRDU intraperitoneally (IP), either naïve (n = 3, white bars) or after two topical R848 treatments (n = 5, grey bars). Mice culled at 16 hr after the BRDU injection and BM harvested. (**E**) Percentage of BRDU positivity in HSPC, MDP, cMoP, GMP, total monocytes, Ly6C-high monocytes, and Ly6C-low monocytes. (**F**) Representative histogram of BRDU expression in HSPC at baseline (blue) or after 2× topical R848 (magenta). (**G**) Percentage of Sca1 positivity in total monocytes, Ly6C-high monocytes and Ly6C-low monocytes in the BM. (**H**) BALB/c mice (n = 3) received four treatments with topical R848. Blood counts for Ly6C-high monocytes (white bars), Ly6C-low monocytes (grey bars), and lymphocytes were monitored at the indicated time points after the cessation of treatment. Data representative of two independent experiments (except **A** and **H**). Two-way ANOVA with Tukey's multiple comparison for time-course experiments (**A–C, H**); unpaired *t*-test (**E, G**). Data are the mean ± SEM; only significant p-values are indicated; *p<0.05; **p<0.01; ***p<0.001; ****p<0.0001.

The online version of this article includes the following source data and figure supplement(s) for figure 4:

**Source data 1.** FACS raw data.

**Figure supplement 1.** Bone marrow (BM) gating strategy.

## Topical R848 accelerates the differentiation of Ly6C-high monocytes to macrophages

Under homeostatic conditions, Ly6C-high monocytes have been shown to be obligate precursors for Ly6C-low monocytes in the blood (*Mildner et al., 2017*). As both BM monocyte subpopulations displayed increased proliferation after topical R848, we questioned whether the Ly6C-low monocytosis under the pathological setting of our experimental model was the result of increased Ly6C-high monocyte conversion or of enhanced egression of Ly6C-low monocytes emerging from the BM as a separate lineage. To answer this question, we utilised an antibody-mediated depletion strategy via injection of an Fc-chimeric mouse anti-GR1. In naïve mice, this antibody fully depletes blood Ly6C-high monocytes by 24 hr post-injection, leaving Ly6C-low cells unaffected (*Figure 5A*). We then administrated this depleting antibody or an anti-FITC isotype control for three consecutive days to R848-treated mice, starting simultaneously with the third R848 treatment at day 6. As expected in untreated mice, the Fc-chimeric mouse anti-GR1 antibody reduced the Ly6C-high monocyte counts to undetectable levels (*Figure 5B*). Importantly, in R848-treated mice the Ly6C-high monocyte depletion due to the Fc-chimeric mouse anti-GR1 antibody abrogated the monocytosis, whilst the isotype control had no effect (*Figure 5B*). These data suggest that the expanded Ly6C-low monocyte population is predominantly, if not entirely, derived from the Ly6C-high monocyte population and not from an expansion of a separate lineage in the BM.

Ly6C-low blood monocytes are viewed as intravascular 'blood macrophages' due to their transcriptional similarity to tissue macrophages (*Mildner et al., 2017*). Therefore, we reasoned that the cutaneous R848 treatment may be driving an accelerated macrophage differentiation program in Ly6C-high monocytes and thereby also promoting macrophage infiltration into tissue. To confirm this, tissue sections from lungs, liver, and kidneys of R848-treated mice were stained with an anti-CD68 antibody. A significant increase of CD68+ cells per field was observed in all these organs (*Figure 5C*). In the kidney, we also used a combination of CD43 and F4/80 staining to distinguish intravascular CD43-high non-classical monocytes from F4/80-high CD43-negative tissue macrophages (*Kuriakose et al., 2019*). Using this strategy, we found that CD43-positive cells were confined to the vasculature of the glomeruli, while F4/80-high macrophages were mainly located in the medulla (*Figure 5D*), findings consistent with the different roles of these two myeloid subpopulations.

Together these data indicate that cutaneous TLR7 activation triggers the egression from the BM of emergency Ly6C-high monocytes that undergo an accelerated macrophage differentiation, both patrolling the vascular lumen as Ly6C-low monocytes and also directly invading different organs as blood-derived tissue macrophages. As the Ly6C-low population remains in the intravascular space, this would explain the progressing switch in the Ly6C-high and Ly6C-low monocyte ratio (*Figure 1B*).

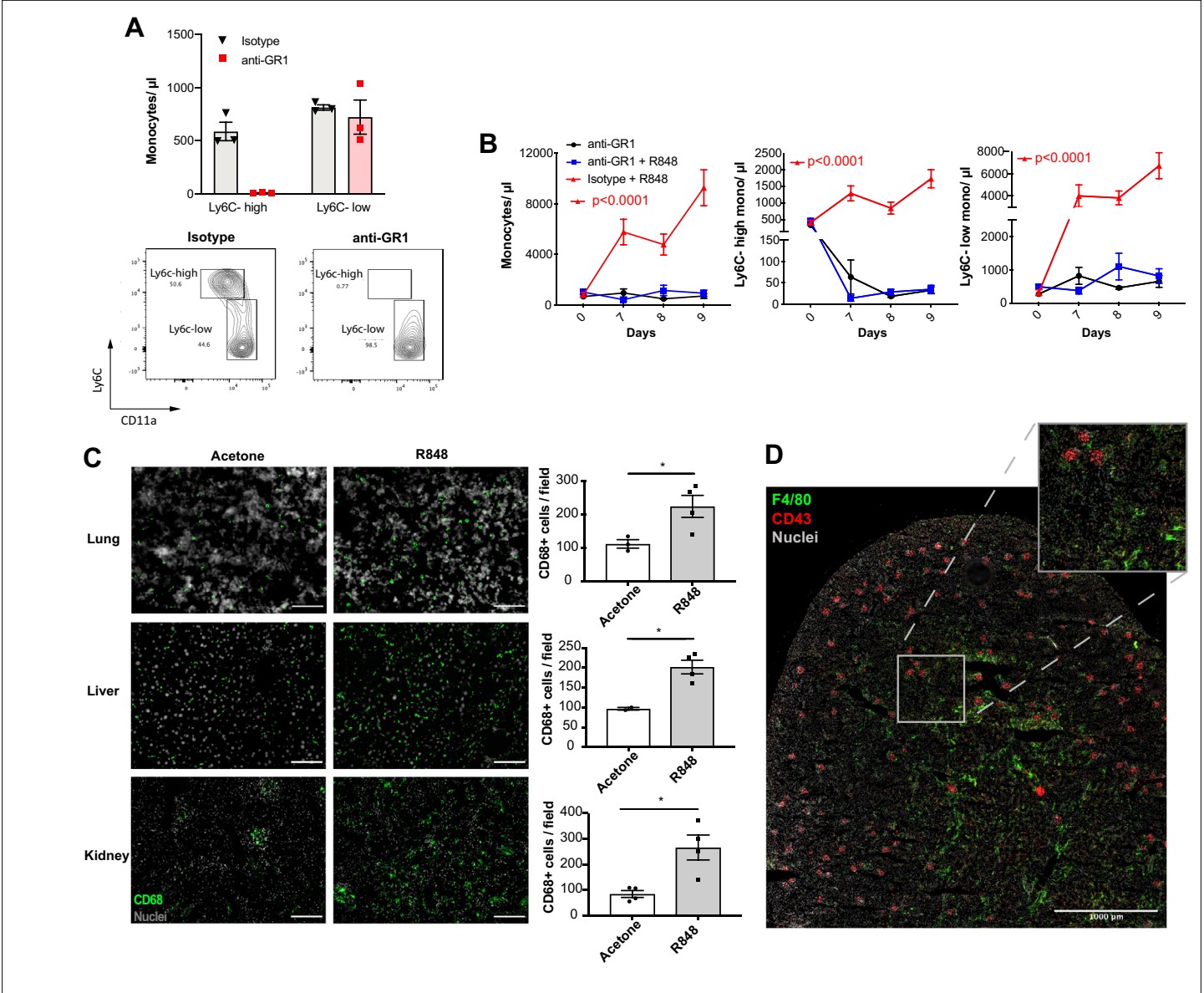

**Figure 5.** Topical R848 drives Ly6C-high monocyte differentiation to intravascular and tissue macrophages. (**A**) C57BL/6 mice were injected intraperitoneally (IP) with either anti-GR1 antibody (n = 3, red squares) or matched isotype control (n = 3, black triangles). Blood Ly6C-high and Ly6C-low monocyte counts (upper panels) and representative flow cytometry plots (bottom panels) are shown at 24 hr post-injection. (**B**) C57BL/6 mice were treated with combinations of topical R848 and IP anti-GR1 antibody or matched isotype control, with the antibody injected daily for three consecutive days starting 1 hr prior to the third R848 treatment (day 6). The groups were isotype control and R848 (n = 4, red triangle), anti-GR1 and R848 (n = 4, blue squares), and anti-GR1 alone (n = 4, black circles). Blood counts for total monocytes, Ly6C-high monocytes, and Ly6C-low monocytes at baseline and on the indicated days are shown. Statistics compare anti-GR1+R848 to isotype +R848. (**C**) C57BL/6 mice received four treatments with topical R848 (n = 4 per group, grey bars) or acetone (n = 4 per group, white bars). Immunofluorescent staining for CD68 (green) and nuclei (grey) was performed on tissue sections from lung, liver, and kidneys. Staining was quantified as mean CD68+ cells per field. (**D**) C57BL/6 mice (n = 4) received six treatments with topical R848. Kidney sections were stained for CD43 (red), F4/80 (green), and nuclei (grey). A representative tile scan is shown. Data representative of two independent experiments (except **D**). Time-course experiments analysed with two-way ANOVA with Tukey's multiple-comparison used to compare between groups at a given time point (**B**); comparison of two groups at a single time point calculated using unpaired *t*-test (**C**). Data are the mean ± SEM; only significant p-values are indicated; *p<0.05.

The online version of this article includes the following source data for figure 5:

**Source data 1.** FACS raw data.

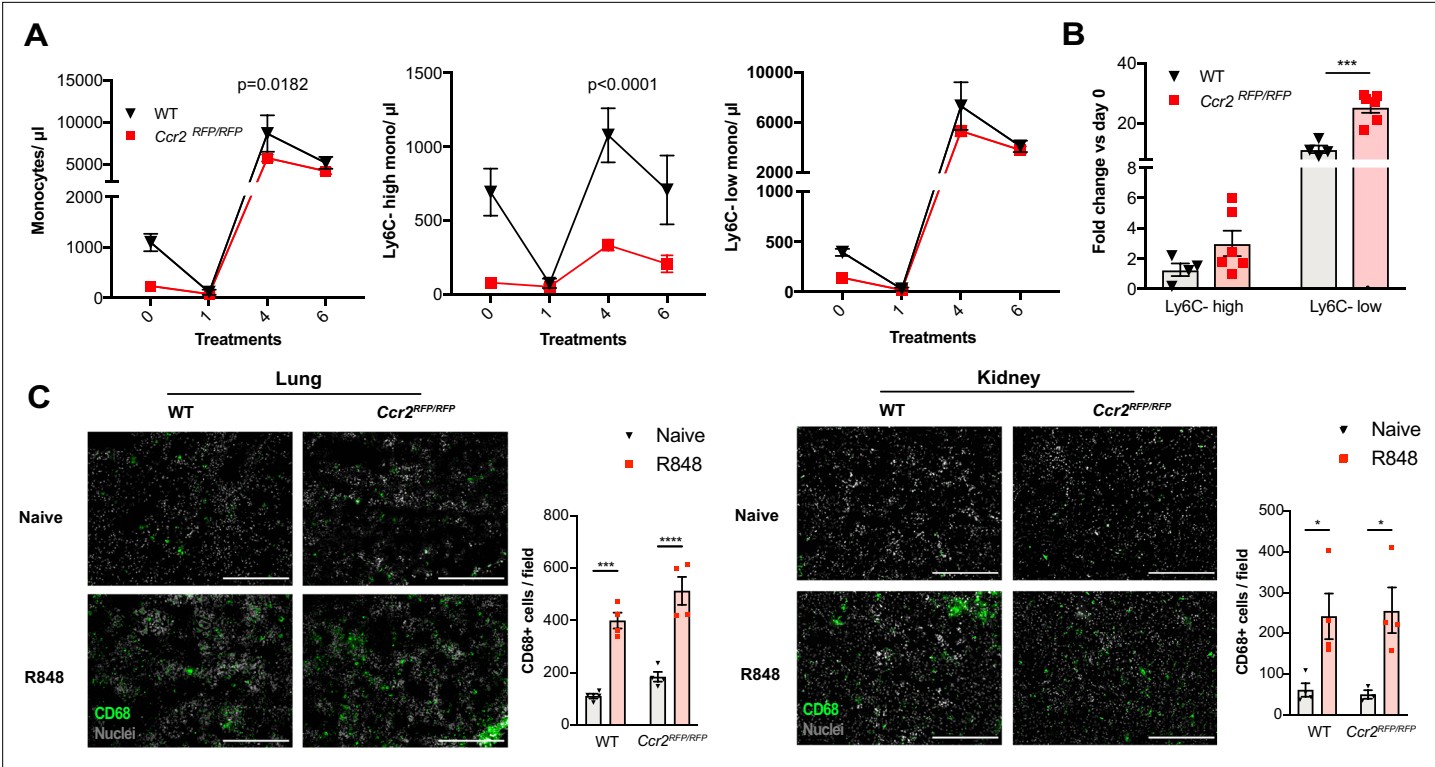

**Figure 6.** Immature monocytes egress from the bone marrow (BM) independently of CCR2. (**A, B**) C57BL/6 mice (n = 4, black triangles) or Ccr2RFP/RFP mice (n = 6, red squares) were treated topically six times with R848. (**A**) Blood counts for total monocytes, Ly6C-high and Ly6C-low monocytes at 24 hr after each treatment are shown. (**B**) Fold change of monocyte subpopulations versus baseline after 6× R848 treatments. (**C**) C57BL/6 mice or Ccr2RFP/RFP mice were either naive (WT n = 4, Ccr2RFP/RFP n = 4, grey bars) or received six R848 treatments (WT n = 4, Ccr2RFP/RFP n = 4, red bars). Immunofluorescent staining for CD68 (green) and nuclei (grey) was performed on tissue sections from lung and kidneys. Staining was quantified as mean CD68+ cells per field. Data representative of two (**C**) or four (**A, B**) independent experiments. Time-course experiments analysed using two-way ANOVA with Tukey's multiple-comparison test to compare between time points (**A**); for a single time point one-way ANOVA with Bonferroni's multiple-comparison test for >2 groups (**B, C**). Data are the mean ± SEM; only significant p-values are indicated; *p<0.05, **p<0.01, ***p<0.001, ****p<0.0001.

The online version of this article includes the following source data and figure supplement(s) for figure 6:

**Source data 1.** FACS raw data.

**Figure supplement 1.** R848-induced myeloid response is independent from CX3CR1 and vascular permeability.

## Topical R848 triggers a CCR2-independent myeloid response

Under steady-state conditions, the monocyte release from the BM is orchestrated by CCR2 and CX3CR1 (*Landsman et al., 2009*; *Tsou et al., 2007*). We investigated whether this was also true for the R848-induced myelopoiesis using *Ccr2RFP/RFP* knock-in mice (*Saederup et al., 2010*). As previously reported (*Tsou et al., 2007*), at baseline CCR2-deficient mice had less blood monocytes than the WT animals due to a failure of Ly6C-high monocytes to egress from the BM (*Figure 6A*). After topical R848 treatments, monocytosis was present in both strains and the Ly6C-low monocyte fold increase from baseline was even higher in the mice lacking CCR2 (*Figure 6B*). Consistent with the notion that the R848-driven monocyte response bypassed the conventional pathways, we found a similar increase of CD68+ cells per field in the kidney and lung of the CCR2-deficient mice, indicating that different signals also orchestrated the monocyte extravasation into the tissues (*Figure 6C*).

We next investigated CX3CR1, which has been shown to regulate peripheral levels of Ly6C-low monocytes (*Landsman et al., 2009*), using the *Cx3cr1GFP/GFP* knock-in mouse. At baseline, in the *Cx3cr1GFP/GFP* animals the peripheral Ly6C-low monocyte levels were approximately 50% of those in WT mice, as previously reported (*Landsman et al., 2009*). However, after 4× topical R848 treatments the *Cx3cr1GFP/GFP* mice developed a monocytosis that was skewed towards Ly6C-low subpopulation as in WT mice (*Figure 6—figure supplement 1*), demonstrating that CX3CR1 is dispensable for R848-driven myelopoiesis. Given the independence of the monocyte egress from prototypical

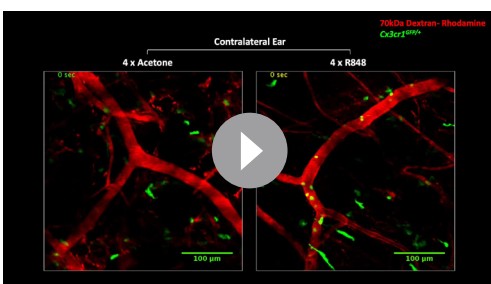

**Video 1.** *Cx3cr1*<sup>GFP/+</sup> mice were treated topically on the right ear for four times with R848 (panel on the right) or acetone (panel on the left). 24 hr after the last treatment, the mice were anaesthetised and the contralateral ear was visualised using intravital microscopy (red = dextran, green = CX3CR1-GFP). https://elifesciences.org/articles/85647/figures#video1

signals, we explored whether topical R848 could be promoting monocyte or haemopoietic progenitor mobilisation in a non-targeted manner by increasing vascular permeability. We injected IV into naïve and R848-treated mice the Evans Blue dye that binds to albumin and would leak into peripheral tissues during situations of decreased vascular integrity (*Radu and Chernoff, 2013*). We found that the Evans Blue amount in the ears from naïve mice was equal to that present in R848-treated ears and the contralateral ears of R848-treated mice (*Figure 6—figure supplement 1*). In addition, there was no increase in BM vascular permeability as assessed by Evans Blue quantity per tibia (*Figure 6—figure supplement 1*). Thus, these findings demonstrate that the cutaneous TLR7 activation was able to stimulate the egression from the BM of monocytes circumventing the requirement of homeostatic cues like CCR2 or CX3CR1 by activating emergency pathways.

## The R848-induced emergency myeloid cells enhance viral control and limit the disease severity

As the BM egression of R848-induced monocytes occurred independently from the conventional regulatory mechanisms, we hypothesised these monocytes could be phenotypically and functionally distinct. Using the *Cx3cr1*<sup>GFP/-</sup> mouse, Ly6C-low monocytes were visualised by intravital microscopy in the contralateral ear of mice treated topically 4× with R848. We observed a dramatic increase (~10-fold) in intravascular CX3CR1-GFP+ monocytes in the R848-treated group (*Figure 7—figure supplement 1*). These monocytes appeared to be rolling on the vessel wall, indicating an activated phenotype (*Video 1*). To confirm this, we comprehensively phenotyped the blood monocytes by flow cytometry. Consistent with the phenotype of the R848-induced bone marrow monocytes (*Figure 4E*), both blood subpopulations displayed expression of Sca1 (*Figure 7A*). In addition, both monocyte subpopulations expressed less F4/80 and more CD115 (*Figure 7B and C*), suggesting an immature phenotype and a recent dependence on M-CSF. These changes were particularly pronounced in the Ly6C-low population, which appeared both activated and immature as reflected by the upregulation of CXCR4, CD11c, CD11a, and CD62L and the downregulation of CCR2 (*Figure 7B*). Most notably, increased CXCR4 and decreased CCR2 expression phenocopy the changes seen in transitional BM pre-monocytes (*Chong et al., 2016*), which are not normally found in the blood, suggesting the premature egress of an emergency population. We next explored whether the R848-induced monocytes were also functionally impaired. We isolated BM monocytes from R848-treated and untreated mice and challenged them in vitro with R848 or LPS. We found that the R848-treated monocytes had a reduced cytokine response when rechallenged with the same TLR7 agonist but not with a different TLR stimulus (*Figure 7—figure supplement 1*), suggesting a pathway-specific adaptation in the monocytes derived from the R848-exposed HSCs.

As the epithelial R848 applications promoted the migration and macrophage differentiation of Ly6C-high monocytes in peripheral organs like the lungs (*Figure 5C*), we next evaluated their contribution to control viral infections. We first infected R848-treated and control mice with the low pathogenicity H3N2 influenza strain X31 (*Davidson et al., 2014*). We found that the animals with the myeloid response showed reduced weight loss compared to untreated mice (*Figure 7D*). Consistent with the change in weight loss, the lung viral load, measured by expression of the influenza X31 matrix gene, was decreased in R848-treated mice (*Figure 7D*). Similarly, the viral replication after infection with the RSV, an RNA virus responsible for infant hospitalisations in the developed world, was limited in R848-treated mice (*Figure 7E*).

Monocytes have previously been shown to have a beneficial antiviral effect in the lungs (*Goritzka et al., 2015*). Together our findings indicate that the extravasation of monocytes into the lungs as

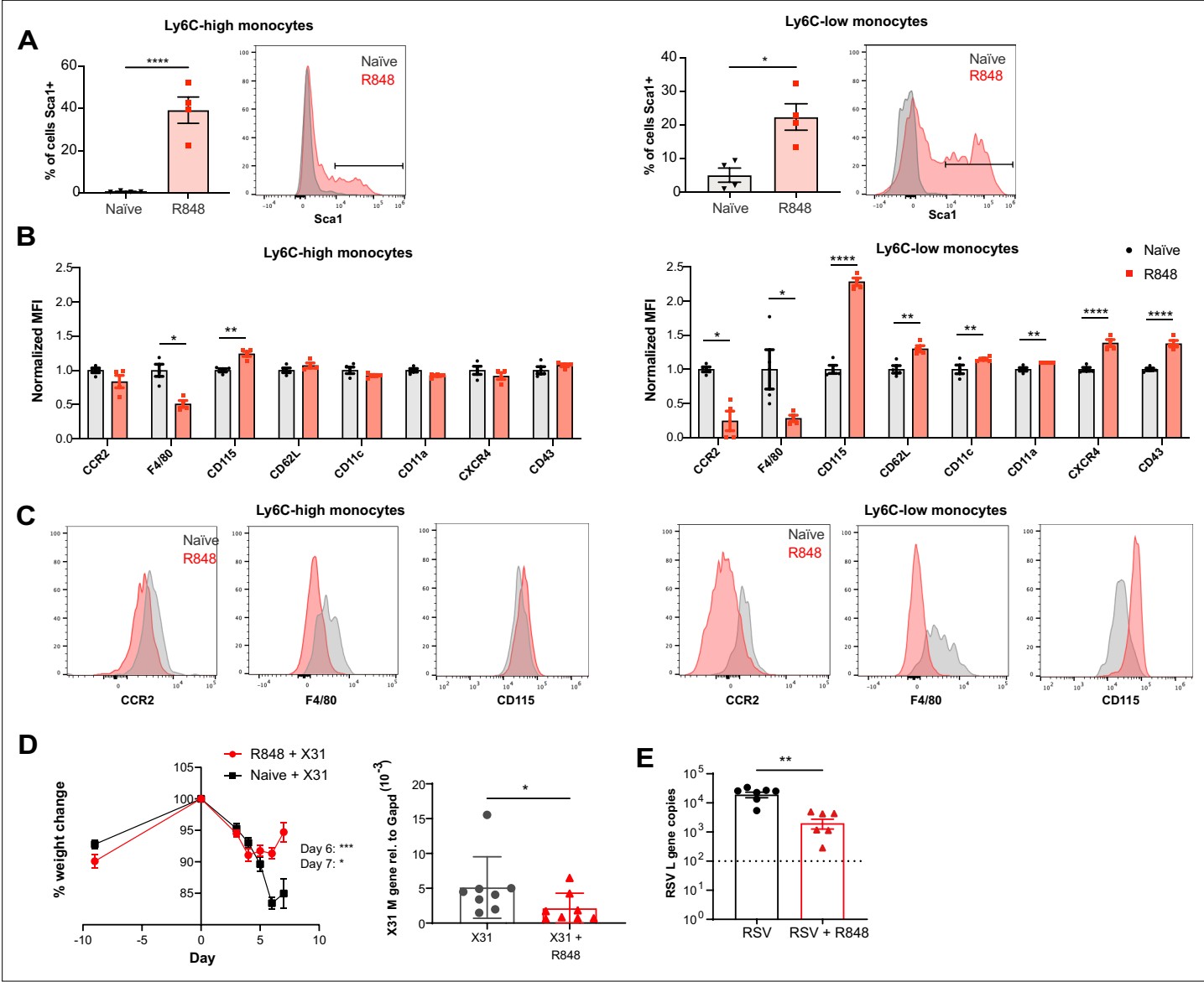

**Figure 7.** R848-induced emergency monocytes have antiviral effects in the lung. (**A–C**) C57BL/6 mice were either naive (n = 4, black circles) or treated four times with topical R848 (n = 4, red squares). (**A**) Percentage of cells positive for Sca-1 and representative histograms in blood Ly6C-high and Ly6C-low monocytes from naive mice (grey) or R848-treated mice (red). (**B**) Blood Ly6C-high and Ly6C-low monocytes were gated by flow cytometry and surface expression of the indicated proteins was quantified and expressed as MFI, normalised to the mean of the naive group. (**C**) Representative histograms of CCR2, F4/80, and CD115 staining in blood Ly6C-high and Ly6C-low monocytes. (**D**) Percentage of original weight (left panel) and lung viral load (right panel) in C57BL/6 mice (n = 8/group) pretreated five times with or without topical R848 infected intranasally with the influenza virus strains X31. Lung viral load at day 7 post inoculation was measured by quantification of matrix X31 gene copies in whole lung tissues. (**E**) Lung viral load after respiratory syncytial virus (RSV) infection in C57BL/6 mice (n = 6–7/group) pretreated or not with topical R848 (five times) was determined by quantification of viral L gene copies in lung tissues at day 4 post inoculation. Data representative of two independent experiments. Statistical analysis using unpaired *t*-test (**A, B; D** right panel, **E**); two-way ANOVA with Bonferroni's post hoc test (**D**, left panel) Data are the mean ± SEM; only significant p-values are indicated; *p<0.05, **p<0.01, ***p<0.001, ****p<0.0001.

The online version of this article includes the following source data and figure supplement(s) for figure 7:

**Source data 1.** FACS raw data.

**Figure supplement 1.** R848-induced monocytes are activated and functionally impaired.

a result of the emergency monopoiesis triggered by cutaneous TLR7 activation is associated with limited viral replication and dampened disease severity. Altogether, this confirms the protective effect mediated by innate immunity even if the encounter with a pathogen-derived product occurs at a distant site.

## Discussion

Increased BM output of inflammatory cells, known as 'emergency myelopoiesis,' is a critical feature of the host response to injury or infection. This process can be driven by systemic inflammatory factors and/or pathogen-derived products acting on precursor cells (*Takizawa et al., 2012*). Herein, we demonstrate that activation of TLR7 at the epithelial barrier of the skin and gut launches a specific and distinct monopoiesis response which does not occur after systemic TLR7 activation or activation of other TLRs. It is characterised by a rapid and dramatic increase of blood monocytes that egress from the BM bypassing the requirement of homeostatic cues like CCR2 or CX3CR1. The TLR7-induced

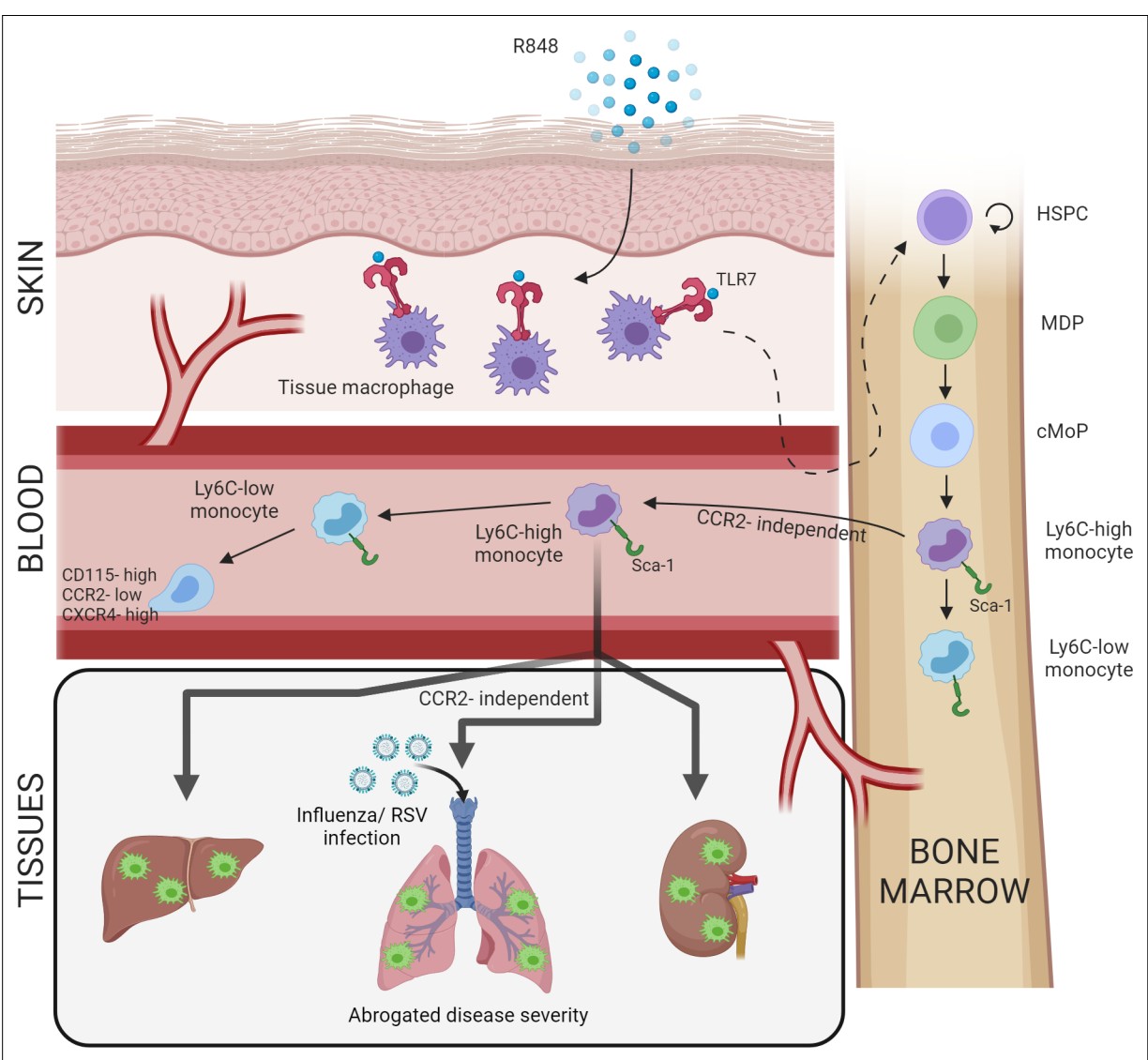

**Figure 8.** Repetitive application of a TLR7 agonist (R848) to the skin activates tissue macrophages which signal to the bone marrow to drive the expansion of haematopoietic stem and progenitor cells (HSPCs) and their differentiation into monocytes. These Ly6C-high monocytes have an immature phenotype and egress from the bone marrow independently of CCR2. Subsequently, Ly6C-high monocytes differentiate into Ly6C-low blood monocytes and into tissue macrophages in multiple organs such as liver, lung, and kidneys. When challenged with a secondary viral stimulus such as influenza or respiratory syncytial virus (RSV) infection, the emergency myelopoiesis is associated with reduced disease severity.

monocytes appear to be immature and programmed to differentiate into macrophages, both surveying the endothelial integrity as Ly6C-low monocytes and trans-migrating into different organs to complement tissue resident macrophages. The response is orchestrated by BM-derived myeloid cells capable of transmitting the signals to the BM HSPCs independently from some of the well-known cytokines such type I and type II IFNs (summarised in *Figure 8*). Collectively our findings define the unique features of the BM myeloid response triggered by single-stranded RNA viruses on entering the epithelial barrier, a condition most likely occurring in infections with arboviruses and coronaviruses.

One of the key observations of our study is the unicity of the initial peripheral signal that is specific to, and shared between, epithelial barriers. Although the detailed mechanisms underlying this specificity remain to be defined, the canonical role of TLR7 in sensing viral ssRNA makes it evolutionarily logical for the response to be specific to epithelial barriers, which must be breached for infection to be established. The skin, lungs, and the gut are similar in their constant exposure to mechanical trauma and a wide range of pathogens. To manage these insults, all these sites are equipped with an immune surveillance network comprising tissue resident macrophages, dendritic cells, and mast cells, as well as resident lymphocytes such as γδ T cells and innate lymphoid cells. Notably, in the skin and gut a subset of macrophages are replenished by circulating monocytes (*Bain et al., 2014*; *Kolter et al., 2019*), unlike in other tissues where macrophages are embryonically derived, proliferate in situ and are only replenished by circulating monocytes after a severe insult (*Ginhoux and Guilliams, 2016*). Interestingly, TLR7 hyperactivation has been shown to drive a myeloid cell expansion in other settings, such as the transgenic overexpression of TLR7 in the TLR7.1 mouse (*Deane et al., 2007*). However, in these models the monocyte response appears to be driven by the extramedullary haematopoiesis in the spleen, whilst in our experimental model the BM is the only source of the myeloid expansion demonstrating the specificity of the peripheral cues induced by TLR7 activation in the skin and gut.

The monocytosis following topical R848 applications is accompanied by a dramatic expansion of HSPCs and monocyte progenitor cells in the BM and the egression in the peripheral blood of transitional BM pre-monocytes. Others have previously described a rapid expansion of phenotypically distinct Ly6C-high monocytes following infections with multiple intestinal pathogens, both parasitic and bacterial (*Abidin et al., 2017*; *Askenase et al., 2015*). In concordance with previous reports, we found that the expanded BM monocyte population expressed high levels of Sca-1 (*Askenase et al., 2015*). However, in our model the induced monocytosis was independent of both type I and type II IFNs, indicating that a different priming mechanism had occurred. One interesting possibility is the type III IFN, which is increasingly recognised as an inducer of local antiviral immunity at epithelial barriers (*Lazear et al., 2015*; *Wack et al., 2015*) and may have a still unrecognised role in the regulation of monocyte progenitors. It is also possible that CSF-1 signalling is involved in the monocyte expansion observed in our experimental model, given the upregulation of the CSF1R (CD115) on the R848-expanded monocytes. Administration of recombinant CSF-1 can expand blood monocytes 5- to 10-fold after 10 d and requires repeated dosing (*Ulich et al., 1990*). However, it remains puzzling why CSF-1 secretion would only be initiated at epithelial barriers and not after systemic activation of TLR7 considering that this can induce extramedullary haematopoiesis in the spleen (*Deane et al., 2007*).

A striking feature of our model is the accelerated differentiation of Ly6C-high to Ly6C-low monocytes. Recent work has suggested that TLR7 activation of Ly6C-high monocytes using topical IMQ or in vitro can directly promote their conversion to Ly6C-low monocytes (*Gamrekelashvili et al., 2020*). However, while we agree that topical TLR7 activation can accelerate monocyte conversion, this appears to be an indirect effect driven by a secondary stimulus, as IV or IP administration of R848 did not result in an obvious monocyte ratio switch. As part of the enhanced differentiation program induced by topical R848, Ly6C-high monocytes also infiltrated peripheral organs and markedly increased the number of tissue macrophages with probably different consequences according to the organ. Global TLR activation using the TLR7.1 transgenic mouse has previously been shown to drive macrophage differentiation to inflammatory hemophagocytic macrophages which infiltrate the spleen and caused anaemia and thrombocytopenia (*Akilesh et al., 2019*). However, unlike the phenotype described here, this process was due to direct TLR7 activation in the BM and could be recapitulated using multiple daily I.P injection of R848, something that we did not detect using a less strong stimulation. In addition, in the paper by *Akilesh et al., 2019* the generation of hemophagocytes could also be promoted by TLR9 activation, whereas we saw no effects of topical CpG application on peripheral monocytes. Whilst it is difficult to reconcile the different findings, one could argue

that the TLR7.1 transgenic mouse and the hyper-stimulation applied by Akilesh et al. in their model may recapitulate extreme phenotypes observed in the macrophage activation syndrome (MAS), a life-threatening complication of rheumatological diseases, whilst our experimental models mimic more closely the BM response to common single-stranded RNA viral infections.

The BM egression of monocytes in steady-steady conditions and during emergency myelopoiesis in response to inflammation and/or infections, including *Toxoplasma gondii* and *Listeria monocytogenes*, has been reported to depend on CCR2 expression (*Grainger et al., 2013*; *Serbina and Pamer, 2006*; *Serbina et al., 2009*) and this has become an accepted dogma in the literature. However, our findings demonstrate unequivocally that topical TLR7 application generates monocytes which are released from the BM in a CCR2-independent process. To our knowledge, the only other context in which this has been shown is during La Crosse virus infection, which interestingly has an ssRNA genome and is detected via TLR7 (*Winkler et al., 2018*). Consistent with a CCR2-independent egress, the R848-induced peripheral blood monocytes have low CCR2 expression and have an immature CXCR4-high, F4/80-low phenotype reminiscent of BM pre-monocytes (*Chong et al., 2016*). Under homeostatic conditions, CXCR4 and CCR2 play antagonistic roles dictating the respective retention or release of Ly6C-high monocytes from the BM (*Jung et al., 2015*). Therefore, the presence of upregulated CXCR4 on blood monocytes potentially indicates their premature release to meet increased demand via a bypass of the CCR2/CXCR4 axis. Of note, a potentially similar phenomenon has been suggested during SARS-CoV-2 infection, based on the presence of proliferation markers on blood monocytes (*Mann et al., 2020*). Given that SARS-CoV-2 has an ssRNA genome, further work is required to determine if our findings are applicable in this setting.

Another striking feature of our model was the accelerated extravasation of the immature R848-induced monocytes into multiple organs. Again, this egression occurred independently from conventional pathways like CCR2/CCL2 and was not associated with any obvious evidence of organ inflammation or damage, suggesting a distinct pathway. Consistent with the notion and previous data (*Goritzka et al., 2015*) that an early recruitment of Ly6C-high monocytes into the lung is a beneficial feature of the host innate immune response to virus infection, we found that the influx of R848-induced emergency myeloid cells restricted RSV and influenza virus replication and dampened disease severity. Similarly, monocyte-derived cells have been shown to limit HSV-2 and HSV-1 replication in the vaginal tract and in the cornea, respectively (*Conrady et al., 2013*; *Iijima et al., 2011*), indicating that these cells play a key role in the resistance to viruses together with the type I IFNs. Notably, the fact that the monocytosis persisted long after the TLR7 stimulation at the epithelial barrier had ceased would suggest that the protective effects provided by these cells may prolong the antiviral effect of the type I IFNs. However, blood-derived tissue macrophages can also induce tissue damage. For example, inflammatory Ly6C-high monocytes have been shown to contribute to lung pathology in a model of influenza virus infection (*Davidson et al., 2014*; *Herold et al., 2008*; *Lin et al., 2008*), most likely recapitulating the pathology that occurs in severe infection if the initial innate response is dysregulated. A similar pattern has also been reported during SARS-CoV-2 infection where dynamic changes in the monocyte response have been correlated with COVID-19 disease severity (*Schulte-Schrepping et al., 2020*). Therefore, recruitment of antiviral monocytes during lung infections must be carefully balanced to reduce viral load but not allow excessive cell infiltration that can cause tissue damage.

In summary, the work presented here elucidates a novel pathway of emergency myelopoiesis which is uniquely activated by TLR7-expressing macrophages at epithelial barriers. Unlike in other settings, this is due to a specific expansion of the monocyte lineage from BM progenitors, is CCR2-independent, and releases atypical pre-monocytes in the periphery. This emergency myeloid population is pushed towards a macrophage differentiation pathway which results in a predominance of Ly6C-low patrolling monocytes in the blood and a simultaneous infiltration of inflammatory macrophages into peripheral organs. Together these data challenge the dogma on how monocytes are produced during viral infections occurring at peripheral sites and highlight their contribution to the initial antiviral immunity.

## Limitations of the study

While this study has revealed a novel pathway of emergency myelopoiesis uniquely triggered by TLR7 activation at epithelial barriers, it has a few limitations. First, we did not identify the mechanism(s) for the remote activation of monocytes. We assume that a soluble factor mediates this effect, but the

nature of this soluble factor remains unclear. Additionally, whilst our study indicates that skin myeloid cells are responsible for this expansion of the monocyte lineage from BM progenitors, we could not pinpoint to a specific skin myeloid cell type. Furthermore, while our experiments demonstrate a CCR2-independent egression of BM immature monocytes as well as CCR2-independent migration of the Ly6C-high monocytes into different organs, we did not define the distinct pathways driving these processes. Moreover, how the R848-induced emergency myeloid cells modulate the lung anti-viral response remains a speculation and further studies will be needed to address these points.

# Materials and methods

## Key resources table

| Reagent type (species) or resource | Designation | Source or reference | Identifiers | Additional information |
|---|---|---|---|---|
| Antibody | Rat anti-mouse Ly6G FITC (1A8) | BioLegend | Cat# 127606 | Flow (1:100) |
| Antibody | Rat anti-mouse MHC-II FITC (M5/114.15.2) | BioLegend | Cat# 107606 | Flow (1:400) |
| Antibody | Rat anti-mouse Ly6C PerCP-Cy5.5 (HK1.4) | BioLegend | Cat# 128012 | Flow (1:200) |
| Antibody | Rat anti-mouse Ly6C Brilliant Violet 605 (HK1.4) | BioLegend | Cat# 128036 | Flow (1:100) |
| Antibody | Rat anti-mouse CD11b PE-Cy7 (M1/70) | BioLegend | Cat# 101216 | Flow (1:400) |
| Antibody | Rat anti-mouse CD11b APC (M1/70) | BioLegend | Cat# 101212 | Flow (1:400) |
| Antibody | Rat anti-mouse CD115 PE (AFS98) | BioLegend | Cat# 135506 | Flow (1:100) |
| Antibody | Rat anti-mouse CD115 APC (AFS98) | BioLegend | Cat# 135510 | Flow (1:100) |
| Antibody | Rat anti-mouse B220 Brilliant Violet 605 (RA3-6B2) | BioLegend | Cat# 103244 | Flow (1:400) |
| Antibody | Rat anti-mouse B220 FITC (RA3-6B2) | BD Biosciences | Cat# 553087 | Flow (1:400) |
| Antibody | Hamster anti-mouse CD3e Brilliant Violet 711 (145-2C11) | BioLegend | Cat# 100349 | Flow (1:100) |
| Antibody | Hamster anti-mouse CD3e FITC (145-2C11) | BD Biosciences | Cat# 553062 | Flow (1:100) |
| Antibody | Rat anti-mouse CD11a APC (M17/4) | BioLegend | Cat# 101120 | Flow (1:100) |
| Antibody | Rat anti-mouse CD117 PE-Cy7 (2B8) | BioLegend | Cat# 105814 | Flow (1:100) |
| Antibody | Rat anti-mouse CD11b PerCP-Cy5.5 (M1/70) | BioLegend | Cat# 101228 | Flow (1:400) |
| Antibody | Rat anti-mouse CD135 Brilliant Violet 421 (A2F10.1) | BioLegend | Cat# 135315 | Flow (1:100) |
| Antibody | Rat anti-mouse Sca1 Brilliant Violet 711 (D7) | BioLegend | Cat# 108131 | Flow (1:100) |
| Antibody | Rat anti-mouse CD16/CD32 APC-Cy7 (93) | BioLegend | Cat# 101328 | Flow (1:100) |
| Antibody | Rat anti-mouse CD135 APC (A2F10.1) | BioLegend | Cat# 135310 | Flow (1:100) |
| Antibody | Rat anti-mouse CD49b FITC (DX5) | BD Biosciences | Cat# 553857 | Flow (1:100) |

*Continued on next page*

*Continued*

| Reagent type (species) or resource | Designation | Source or reference | Identifiers | Additional information |
|---|---|---|---|---|
| Antibody | Hamster anti-mouse CD11c APC (HL3) | BD Biosciences | Cat# 550261 | Flow (1:100) |
| Antibody | Anti-mouse CD45 APC-eFluor 780 (30-F11) | eBioscience | Cat# 47-0451-82 | Flow (1:100) |
| Antibody | Rat anti-mouse Ly6G Brilliant Violet 421 (1A8) | BioLegend | Cat# 127628 | Flow (1:100) |
| Antibody | Rat anti-mouse F4/80 Brilliant Violet 785 (BM8) | BioLegend | Cat# 123141 | Flow (1:400) |
| Antibody | Rat anti-mouse F4/80 Alexa Fluor 488 (BM8) | BioLegend | Cat# 123120 | IF (1:200) |
| Antibody | Rat anti-mouse MHC-II PerCP eFluor710 (M5/114.15.2) | eBioscience | Cat# 46-5321-82 | Flow (1:400) |
| Antibody | Rat anti-mouse Sca1 PE-Dazzle 594 (D7) | BioLegend | Cat# 108138 | Flow (1:100) |
| Antibody | Rat anti-mouse CD43 Alexa Fluor 700 (S11) | BioLegend | Cat# 143214 | Flow (1:400) |
| Antibody | Rat anti-mouse CD43 | BioLegend | Cat# 143202 | IF (1:50) |
| Antibody | Goat anti-Art IgG (H+L) Cross-Adsorbed secondary antibody, Alexa-Fluor 555 | Thermo Fisher | Cat# A-21434 | IF (1:400) |
| Antibody | Rat anti-mouse CXCR4 Alexa Fluor 647 (L276F12) | BioLegend | Cat# 146504 | Flow (1:100) |
| Antibody | Rat anti-mouse CCR2 Brilliant Violet 510 (SA203G11) | BioLegend | Cat# 150617 | Flow (1:100) |
| Antibody | Hamster anti-mouse CD11c PE-Cy7 (HL3) | BD Biosciences | Cat# 561022 | Flow (1:100) |
| Antibody | Rat anti-mouse Ly6C eFluor450 (HK1.4) | eBioscience | Cat# 48-5932-82 | Flow (1:100) |
| Antibody | Rat anti-mouse CD68 Alexa Fluor 488 (FA-11) | BioLegend | Cat# 137012 | IF (1:100) |
| Antibody | Rat anti-mouse Ly6G eFluor450 (1A8) | eBioscience | Cat# 48-9668-82 | Flow (1:100) |
| Antibody | Rat anti-mouse B220 PE-Cy5 (RA3-6B2) | BioLegend | Cat# 103210 | Flow (1:400) |
| Antibody | Rat anti-mouse CD4 PE-Cy5 (RM4-5) | BioLegend | Cat# 100513 | Flow (1:100) |
| Antibody | Rat anti-mouse CD5 PE-Cy5 (53-7.3) | BioLegend | Cat# 100604 | Flow (1:100) |
| Antibody | Rat anti-mouse CD8 PE-Cy5 (53-6.7) | BioLegend | Cat# 100710 | Flow (1:100) |
| Antibody | Rat anti-mouse TER119 PE-Cy5 (TER119) | BioLegend | Cat# 116210 | Flow (1:100) |
| Antibody | Rat anti-mouse GR1 PE-Cy5 (RB6-8C5) | BioLegend | Cat# 108410 | Flow (1:100) |
| Antibody | Rat anti-mouse CD150 PE-Cy7 (TC15-12F12.2) | BioLegend | Cat# 115914 | Flow (1:100) |
| Antibody | Rat anti-mouse CD117 APC eFluro780 (2B8) | eBioscience | Cat# 47-1171-82 | Flow (1:100) |

*Continued on next page*

*Continued*

| Reagent type (species) or resource | Designation | Source or reference | Identifiers | Additional information |
|---|---|---|---|---|
| Antibody | Rat anti-mouse Sca1 BV785 (D7) | BioLegend | Cat# 108139 | Flow (1:100) |
| Antibody | Hamster anti-mouse CD48 APC (HM48-1) | BioLegend | Cat# 103412 | Flow (1:100) |
| Antibody | Anti-mouse Ly6G (1A8, mouse chimeric) | Absolute Antibody | Cat# Ab00295-2.3 | Flow (1:100) |
| Antibody | Anti-mouse GR1 (RB6-8C5, mouse chimeric) | Absolute Antibody | Cat# Ab01030-2.0 | Flow (1:100) |
| Antibody | Rat anti-mouse IL-6R (15A7) | BioXcell | Cat# BE0047 | |
| Antibody | Rat IgG2b isotype control, anti-keyhole limpet hemocyanin (LTF-2) | BioXcell | Cat# BE0090 | |
| Peptide, , recombinant protein | R848 (water soluble) | Invivogen | tlrl-r848 | |
| Peptide, , recombinant protein | R848 | Enzo | ALX-420-038M025 | |
| Peptide, , recombinant protein | Poly(I:C) (LMW) | Invivogen | tlrl-picw | |
| Peptide, , recombinant protein | CpG (ODN 2395) | Invivogen | tlrl-2395 | |
| Peptide, , recombinant protein | LPS (*E. coli* 055:B5) | Invivogen | tlrl-b5lps | |
| Chemical compound, drug | Aldara (Imiquimod) | Meda Pharmaceuticals | | 5% cream |
| Chemical compound, drug | Anakinra | Swedish Orphan Biovitrum | | 150 mg/ml |
| Chemical compound, drug | Etanercept (Enbrel) | Pfizer Europe | | 25 mg |
| Chemical compound, drug | Baytril | Bayer Corporation | | |
| Chemical compound, drug | Tamoxifen | Sigma | Cat# T5648-1G | |
| Chemical compound, drug | Phorbol 12-myristate 13-acetate (TPA) | Sigma-Aldrich | Cat# P8139 | |
| Chemical compound, drug | Tetramethylrhodamine isothiocyanate–Dextran, 70 kDa | Sigma-Aldrich | Cat# T11-62 | |
| Chemical compound, drug | Bromodeoxyuridine (BrDU) | BioLegend | Cat# 423401 | |
| Chemical compound, drug | Liberase TM Research grade | Roche | Cat# 5401119001 | |
| Chemical compound, drug | DNase I (grade II) from bovine pancreas | Sigma-Aldrich | Cat# 10104159001 | |
| Commercial assay or kit | RNA-Later | Thermo Fisher | Cat# AM7020 | |
| Commercial assay or kit | LIVE/DEAD Fixable Aqua Dead Cell Stain Kit | Life Technologies | Cat# L34597 | |
| Commercial assay or kit | BD FACS Lysis Solution 10X Concentrate | BD Biosciences | Cat# 349202 | |

*Continued on next page*

*Continued*

| Reagent type (species) or resource | Designation | Source or reference | Identifiers | Additional information |
|---|---|---|---|---|
| Commercial assay or kit | RNEasy Mini Kit | QIAGEN | Cat# 74104 | |
| Commercial assay or kit | RNEasy Micro Plus Kit | QIAGEN | Cat# 74034 | |
| Commercial assay or kit | iScript cDNA Synthesis Kit | Bio-Rad | Cat# 1708891 | |
| Commercial assay or kit | QuantiTect Probe PCR Kit | QIAGEN | Cat# 204343 | |
| Commercial assay or kit | High-Capacity RNA-to-cDNA kit | Applied Biosystems | Cat# 4387406 | |
| Commercial assay or kit | Legendplex Mix and Match Kit | BioLegend | | |
| Commercial assay or kit | BrdU Staining Kit | eBioscience | Cat# 8817-6600 | |
| Strain, strain background (*Mus musculus*, C57BL/6) | B6.129P-Cx3cr1$^{tm1Litt}$/J | Jackson Laboratory | JAX stock #005582 | *Jung et al., 2000* |
| Strain, strain background (*M. musculus*, C57BL/6) | B6(Cg)-Ifnar1tm1.2Ees/J | Jackson Laboratory | JAX stock #028288 | *Hwang et al., 1995* |
| Strain, strain background (*M. musculus*, C57BL/6) | B6(Cg)-Rag2$^{tm1.1Cgn}$/J | Jackson Laboratory | JAX stock #08449 | *Hao and Rajewsky, 2001* |
| Strain, strain background (*M. musculus*, C57BL/6) | B6.129P2-Lyz2$^{tm1(cre)Ifo}$/J | Jackson Laboratory | JAX stock #004781 | *Clausen et al., 1999* |
| Strain, strain background (*M. musculus*, C57BL/6) | B6.129(Cg)-Ccr2$^{tm2.1Ifc}$/J | Jackson Laboratory | JAX stock #017586 | *Saederup et al., 2010* |
| Strain, strain background (*M. musculus*, C57BL/6) | B6.129P2-Tlr7$^{tm1Aki}$ | *Hemmi et al., 2002* | | |
| Strain, strain background (*M. musculus*, C57BL/6) | B6.129S7-Ifng$^{tm1Ts}$/J | Jackson Laboratory | JAX stock #002287 | *Dalton et al., 1993* |
| Strain, strain background (*M. musculus;* C57BL/6) | Tlr7$^{flox/flox}$ | *Solmaz et al., 2019* | | |
| Strain, strain background (*M. musculus;* BALB/c) | Cpa3- cre$^{4Glli}$ | Jackson Laboratory | JAX stock #026828 | *Feyerabend et al., 2011* |
| Strain, strain background (*M. musculus;* BALB/c) | Gata1$^{tm6Sho}$/J | Jackson Laboratory | JAX stock #05653 | *Yu et al., 2002* |
| Strain, strain background (*M. musculus;* C57BL/6) | HSC-SCL-Cre-ER$^{T}$;R26R-EYFP | *Göthert et al., 2005* | | |
| Strain, strain background (influenza A virus) | strain X31 | John McCauley | | Davidson, S., 2014 |

*Continued on next page*

*Continued*

| Reagent type (species) or resource | Designation | Source or reference | Identifiers | Additional information |
|---|---|---|---|---|
| Strain, strain background (RSV) | Strain A2 | ATCC | ATCC VR-1540 | |
| Sequence-based reagent | RSV L gene forward | Invitrogen | | GAACTCAGT GTA GGT AGAATGTTTGCA |
| Sequence-based reagent | RSV L gene reverse | Invitrogen | | TTCAGCTATCATTTTCTCTGCCAAT |
| Sequence-based reagent | RSV L FAM-TAMRA probe | Eurofins MWG Operon | | TTTGAACCTGTCTGAACATTCCCGGTT |
| Sequence-based reagent | Flu M1 gene forward | Invitrogen | | AAGACCAATCCTGTCACCTCTGA |
| Sequence-based reagent | Flu M1 gene reverse | Invitrogen | | CAAAGCGTCTACGCTGCA |
| Sequence-based reagent | Flu M1 FAM-TAMRA probe | Eurofins MWG Operon | | TTTGTGTTCACGCTCACCGT |
| Software, algorithm | GraphPad Software (Prism) | GraphPad Software, Inc, La Jolla, California, USA | Version 9 | |
| Software, algorithm | FlowJo | Tree Star Inc Ashland, OR, USA | Version 10.7.1 | |
| Software, algorithm | Imaris 8.0.1 | Bitplane AG | | |
| Software, algorithm | FIJI | ImageJ2 (open source) | | |
| Software, algorithm | 7500 Fast System SDS v1.4 21 CFR Part 11 Module | Applied Biosystems | | |
| Software, algorithm | QuantStudio Software V1.2.4 | Applied Biosystems | | |

## Mouse strains

All procedures were carried out in accordance with the institutional guidelines and the studies were approved by the UK Home Office. Experimental studies were designed according to the ARRIVE guidelines (*Percie du Sert et al., 2020*). Experimental mice were 8–12 wk of age, sex-, and age-matched. All animals were housed in individually ventilated cages. The animals were selected randomly from a large pool, but no specific method of randomisation was used to allocate mice into groups. The investigators were not blinded to allocation during experiment and outcome assessment.

BALB/c and C57BL/6 mice were purchased from Charles River (UK). The following mice were as previously described and maintained on a C57BL/6 background: B6.129P-*Cx3cr1*[tm1Litt/J](*Cx₃cr1*[GFP/GFP]) (*Jung et al., 2000*), Ifnar1[-/-] (*Hwang et al., 1995*), B6(Cg)-Rag2[tm1.1Cgn]/J (*Rag2*[-/-]) (*Hao and Rajewsky, 2001*), B6.129P2-Lyz2[tm1(cre)Ifo]/J (*Lyz2*[Cre]) (*Clausen et al., 1999*), B6.129(Cg)-Ccr2[tm2.1Ifc]/J (*Ccr2*[RFP/RFP]) (*Saederup et al., 2010*), B6.129P2-Tlr7[tm1Aki] (*Tlr7*[-/-]) (*Hemmi et al., 2002*), B6.129S7-[Ifngtm1Ts]/J (*Ifng*[-/-]) (*Dalton et al., 1993*), *Tlr7*[fl] (*Solmaz et al., 2019*), and HSC-SCL-Cre-ER[T];R26R-EYFP (*Göthert et al., 2005*). The following mice were as previously described and maintained on a BALB/c background: *Gata1*[tm6Sho]/J (ΔdblGATA) (*Yu et al., 2002*), Cpa3[Cre/+] (*Feyerabend et al., 2011*).

## In vivo administration of TLR agonists

### Topical application

Mice were treated on the dorsal side of the right ear with 100 µg of R848 (Enzo Life Sciences, USA) diluted in 30 µl of acetone, three times per week up to a maximum of 12 treatments. The same dosing regimen was used for the following TLR ligands: lipopolysaccharide (LPS) from *Escherichia coli* 055:B5 (Invivogen), low molecular weight polyinosine-polycytidylic acid (Poly(I:C), Invivogen), Class C CpG oligonucleotide ODN-2395 (Invivogen). They were dissolved in 50:50 DMSO:PBS and applied topically at 100 µg per dose. Imiquimod (IMQ) at a dose of 5 mg of 5% IMQ cream (Aldara cream, 3M

Pharmaceutical) was applied on the ventral side of one ear for five consecutive days. 5 nmol of 12-O
-tetradecanoylphorbol-13-acetate (TPA) dissolved in ethanol was applied topically on the dorsal ear
skin four times over 2 wk.

### Other routes of administration

(i) Mice treated intraperitoneally IP received 100 µg of a water-soluble R848 formulation (Invivogen)
dissolved in 200 µl of PBS, three times per week; (ii) mice treated intravenously received 100 µg of
water-soluble R848 in 100 µl of PBS, three times per week. Vehicle-treated mice were used as controls;
(iii) mice treated orally received drinking water supplemented with R848 (Invivogen) at 6.3 µg/ml to
provide a calculated dose of ~50 µg/day/mouse. Sodium saccharine sweetener (Sweetex) was added
at two tablets per 100 ml to improve palatability. Monocyte counts were monitored by flow cytometry
of the peripheral blood at the indicated time points, as described in blood sampling.

## Blocking and depletion experiments

### Blocking antibody

Mice were injected IP with 10 mg of the anti-IL1R Anakinra (Kineret, Sobi, Sweden) for seven consec-
utive days, starting on the day of the first R848 treatment. Mice simultaneously received topical treat-
ments every other day with either R848 or acetone. Mice were injected IP with 200 mg of the anti-TNF
Etanercept (Enbrel) or PBS for nine consecutive days, starting 2 d prior to the first R848 treatment.
Mice were injected IP with 200 mg of the anti-IL-6R (Clone 15A7, BioXcell) or isotype control (BioX-
cell) or PBS every 3 d, starting 1 d prior to the first R848 treatment. Mice then received topical R848
treatments every other day for four times. Monocyte counts were assessed by flow cytometry at the
indicated time points, as described in blood sampling.

### Neutrophil depletion

Mice were injected IP with 200 µg of a chimeric anti-Ly6G antibody composed of the 1A8 variable
region with a mouse IgG2a Fc region to avoid eliciting a neutralising response (Absolute Antibody)
(*Daley et al., 2008*). Mice received four topical R848 treatments at a frequency of three treatments
per week and antibody was injected 4 hr prior to each R848 treatment. Blood neutrophil depletion
was validated by flow cytometry, with neutrophils defined independently of their Ly6G expression as
$SSC^{high}$, $CD11b^{pos}$, $Ly6C^{int}$, $CD115^{neg}$.

### Ly6C-high monocyte depletion

Mice were injected IP with 100 µg of a chimeric anti-GR1 antibody composed of the RB6-8C5 variable
region with a mouse IgG2a Fc region to avoid eliciting a neutralising response (Absolute Antibody).
Mice received four topical R848 treatments at a frequency of three treatments per week and anti-
body was injected on three consecutive days, starting with the third R848 treatment. Blood monocyte
depletion was validated by flow cytometry at 24 hr after each antibody injection.

## Splenectomy

Eight-week-old BALB/c mice underwent surgical splenectomy or sham surgery under aseptic condi-
tions. Briefly, mice were injected subcutaneously with ketamine (80 mg/kg) and xylazine (16 mg/kg)
before being anaesthetised with 5% isofluorane. Mice were rested for 6 wk before being treated six
times with 100 µg topical R848 over 2 wk. To assess monocytosis, a blood sample was taken prior to
the first R848 treatment and at 24 hr after each treatment.

## Intravital microscopy

$Cx_3cr1^{GFP/+}$ mice, 10–12 wk of age, were treated topically for four times with R848 or acetone on
one ear, before being used for imaging. Intravital imaging of the dermis of the contralateral ear was
performed as previously described (*Carlin et al., 2013*). Briefly, after IP administration of anaesthetic
cocktail of fentanyl/fluanisone and midazolam, mice were maintained at 37°C with oxygen supple-
mentation. 80 µl of tetramethylrhodamine (TRITC) conjugated 70 kDa dextran (70 µM) was injected IV
and the ear to be imaged was taped to the centre of the coverslip. Light was generated from 488 nm
and 562 nm lasers, emitted light signal was detected to generate two colour 8-bit images using a

×10/0.4 objective on a Leica SP5 confocal microscope. Images were analysed using Imaris software (Bitplane). Dextran signal was used to identify the intravascular cells and monocytes were automatically selected on the base of the quality and intensity of their GFP signal.

## BrDU incorporation

C57BL/6 mice were treated twice with topical R848 and then were injected once IP with 2 mg of (5-bromo-2'-deoxyuridine) BrdU (BioLegend). BM was harvested 16 hr later and samples were processed with BrdU staining kit (eBioscience, USA) according to the manufacturer's provided protocol and analysed by flow cytometry (see details in 'Materials and methods').

## Lineage tracing experiment

HSC-SCL-Cre-ER$^T$;R26R-EYFP mice were given 100 µl tamoxifen (40 mg/ml) by oral gavage for five consecutive days. After 3 d, some mice received topical R848 treatments every other day for four times and others were left untreated. Monocyte counts were assessed by flow cytometry at the indicated time points, as described in blood sampling.

## Bone marrow chimeras

Prior to irradiation, all mice were treated orally with Baytril (Bayer) for 7 d and moved to sterile cages. C57BL/6 or *Tlr7*$^{-/-}$ host mice were lethally irradiated (700 cGy) and reconstituted IV on the same day with 5 × 10$^6$ BM cells from either C57BL/6 or *Tlr7*$^{-/-}$ donor mice. Mice were left to recover for 8 wk and reconstitution was confirmed by blood sampling and flow cytometry. Reconstituted mice were treated topically with 100 µg R848, three times per week for four treatments.

## Vascular permeability

BALB/c mice were naive or treated topically four times with 100 µg R848. 200 µl of 0.5% Evans Blue in PBS was injected IV at 24 hr after the final R848 treatment. Mice were culled by cervical dislocation after 30 min, and ears and right tibia were harvested. Ears were minced into a slurry and tibias were crushed, before adding each to 500 µl formamide. After 48 hr of incubation at room temperature, the absorbance of the formamide was read at 620 nm and concentration of Evans Blue was calculated using a standard curve.

## Virus infections and qPCR

C57BL/6 were naive or treated topically five times with 100 µg R848 prior to infection with the influenza A virus strain X31 (obtained from John McCauley, The Francis Crick Institute, UK) or plaque-purified human RSV (originally A2 strain from ATCC, USA, grown in HEp2 cells). The topical application of R848 was continued during the viral infection. For infections, mice were lightly anaesthetised (3% isofluorane) and virus was intranasally (IN) administered. Mice received 8 × 10$^5$ focus-forming units (FFU) of RSV or 250 plaque-forming units (PFU) of X31, both given in 100 µl. Weight was monitored daily. Mice were sacrificed at day 7 (X31) and day 4 (RSV) post infection (p.i.) and lung lobes were stored in RNA-later (Sigma). Lung tissue was homogenised using a TissueLyser LT (QIAGEN), and total RNA was extracted using RNeasy Mini kit including DNA removal (QIAGEN) according to the manufacturer's instructions. 1–2 µg of RNA was reverse-transcribed using the High-Capacity RNA-to-cDNA kit according to the manufacturer's instructions (Applied Biosystems). qPCR was performed to quantify lung RNA levels using the mastermix QuantiTech Probe PCR kit (QIAGEN). To quantify RSV L genes, primers and FAM-TAMRA probes previously described were used (*Lee et al., 2010*). The absolute number of gene copies was calculated using a plasmid DNA standard curve and the results were normalised to levels of *Gapdh* (Applied Biosystems). The relative quantification of X31 M genes (primers from *Ward et al., 2004*) was expressed relatively to the expression of *Gapdh*. First, the ΔCt (Ct = cycle threshold) between the target gene and the *Gapdh* for each sample was calculated, then the expression was calculated as $2^{-\Delta Ct}$. Analysis was performed using 7500 Fast System SDS software (Applied Biosystems).

## In vitro stimulation

C57BL/6 mice were naïve or treated topically four times with 100 µg R848. Femur and tibia were harvested without breaking the bone, to maintain sterility. Collected bones were placed into RPMI

1640 (Gibco), 10% FBS, 100 U/ml penicillin, and 100 g/ml streptomycin. Bones were cleaned and flushed with HBSS- 2% FCS and cells were pooled from two mice per biological replicate. Cell suspensions were washed and re-suspended in hypotonic red blood cell lysis buffer (RBC Lysis Buffer: 1 l of water, 8.02 g NH$_4$Cl, 0.84 g NaHCO$_3$, 0.37 g EDTA) for 5 min on ice. Cells were wash and resuspended in PBS solution containing 1% BSA, 2 mM EDTA (Sorting buffer), and cell surface staining was then performed using the fluorochrome-conjugated antibodies as in 'Flow cytometry'. Monocytes were sorted into HBSS- 2% FCS in 15 ml Falcon tubes using a BD Aria III (BD Biosciences). Monocytes were gated as Lineage-negative (B220, CD49b, Ly6G), CD115-positive, CD117-negative, and CD11b-positive. Sorted cells were washed and resuspended in DMEM (Gibco), 20% FCS, 100 U/ml penicillin, and 100 g/ml streptomycin, 2 mM L-glutamine at $1 \times 10^6$ per ml. $1 \times 10^5$ cells per replicate were stimulated with either 100 ng/ml R848 (Invivogen) or 1 μg/ml LPS (O127:B8, Sigma) or medium alone for 16 hr. Supernatant was harvested and cytokines were quantified using a Legendplex Mix and Match Kit according to the manufacturer's instructions (BioLegend).

## Method details

### Flow cytometry

Where live/dead staining was performed, single-cell suspensions were stained for 20 min at room temperature with Live/Dead Fixable Dead Cell Stain (Life) at the dilution of 1/1000 in PBS, as per the manufacturer's protocol. Cells were incubated with a saturating concentration of 2.4G2 monoclonal antibody (anti-CD16/32) to block non-specific Fc receptor binding and stained in PBS solution containing 1% BSA, 2 mM EDTA, and 0.09% NaN$_3$ (FACS buffer) with an appropriate dilution of fluorophore-conjugated antibodies for 20 min at 4°C (see antibodies used in the Key Resources Table). Samples were acquired on either a Fortessa X20 (BD Biosciences) or an Aurora (Cytek) flow cytometer and analysed using FlowJo X for Mac (TreeStar).

### Blood sampling

Tail vein blood samples were added to an equal volume of 100 mM EDTA, washed, and resuspended in FACS buffer. Cell surface staining was then performed using the fluorochrome-conjugated antibodies listed in 'Flow cytometry'. After antibody staining, cells were washed and red blood cells were removed by hypotonic lysis using BD FACS Lysing Solution (BD Biosciences). Absolute cell counts were quantified with AccuCheck counting beads (Thermo Fisher Scientific, USA) as per the manufacturer's protocol. Total blood monocytes were gated as follows and as illustrated in *Figure 1C*: CD3-neg, B220-neg, CD11b+, CD115+. Monocytes were then split into Ly6C-high and Ly6C-low subpopulations based on their expression of Ly6C and CD11a.

### Bone marrow staining

Femur and tibia were harvested without breaking the bone to avoid blood contamination. Collected bones were placed into RPMI 1640 (Gibco), 10% FBS, 100 U/ml penicillin, and 100 g/ml streptomycin. Bones were cleaned and flushed with HBSS- 2% FCS, and cell suspensions were washed and resuspended in hypotonic BD FACS Lysing Solution (BD Biosciences) for 2 min on ice. Cells were washed and resuspended in FACS buffer. Antibody staining was performed as described above in 'Flow cytometry'. The gating for BM populations was adapted from a published strategy (*Yáñez et al., 2017*) and is illustrated in *Figure 4—figure supplement 1*. All populations are first gated lineage-negative (CD3, B220, Ly6G, CD49b). Haematopoietic stem and progenitor cells, HSPC: CD115-neg, c-kit+, Sca-1-neg; monocyte-dendritic cell progenitor, MDP: CD115+, c-Kit+, Ly6C-low; common monocyte progenitor, cMoP: CD115+, c-Kit+, Ly6C-high, Flt3-neg; granulocyte-monocyte progenitor, GMP: CD115-neg, cKit+, Sca-1-neg, CD16/32 + , Ly6C-low; common myeloid progenitor, CMP: CD115-neg, cKit+, Sca-1-neg, CD16/32-neg, Ly6C-low; granulocyte-committed progenitor, GP: CD115-neg, cKit+, Sca-1-neg, CD16/32+, Ly6C-high; Ly6C-high monocytes: CD115+, c-Kit-neg, Ly6C-high; Ly6C-low monocytes: CD115+, c-Kit-neg, Ly6C-low.

### Skin digestion

Single-cell suspensions from ear skin was obtained as follows: (i) ears were split dorsal-ventral using forceps to analyse only the dorsal (treated) side; (ii) skin was cut into ~1 mm$^2$ pieces using a scalpel

blade and incubated for 2 hr in digestion buffer containing 25 µg/ml Liberase (Roche), 250 µg/ml DNAseI (Roche), and 1×DNAse buffer (1.21 Tris base, 0.5 g $MgCl_2$ and 0.073 g $CaCl_2$) at 37°C; (iii) pieces were transferred into C-tubes (Miltenyi Biotech) containing RPMI-1640 medium (Thermo Fisher) supplemented with 10% heat-inactivated FCS and physically disrupted using a GentleMACS dissociator (Miltenyi); (iv) cell suspensions were then filtered through 70 µM cell strainers (BD Biosciences) and counted using a CASY cell counter (Roche). Cell suspensions were stained as described above in 'Flow cytometry'.

### Immunohistochemistry

Lungs, livers, and kidneys were harvested and snap frozen in OCT using isopentane cooled to –80°C on dry ice. 9 µM sections were cut using a Leica JUNG CM1800 cryostat and stored at –80°C. For staining, slides were equilibrated to room temperature before fixation in ice-cold acetone for 10 min. Sections were blocked in PBS with 5% BSA and 10% serum (dependent on the species of the secondary antibody being used) for 1 hr at room temperature. All antibody staining steps were performed in PBS 5% BSA 0.1% Triton X-100 for 45 min at room temperature. The following primary antibodies were used: anti-CD68 Alexa Fluor 488 (FA-11, BioLegend), anti-F4/80 Alexa Fluor 488 (BM8), anti-CD43 (S11, BioLegend). Hoechst 33342 (NucBlue, Thermo Fisher) was added to the final staining step according to the manufacturer's instructions and sections were mounted in ProLong Glass (Thermo Fisher). Images were acquired on a Zeiss Axio Observer inverted widefield microscope using a Colibri.2 LED illumination source, a ×20/0.8 plan-apochromat objective and a Hamamatsu Flash 4.0 camera. Images were processed to correct brightness and contrast in FIJI (*Schindelin et al., 2012*). Cell quantification was performed using the surfaces function of Imaris 8 (Bitplane), with a minimum of five randomly selected fields per section being used for analysis. Data are expressed as the mean number of cells per field.

## Quantification and statistical analysis

Statistical comparisons between two groups at a single time point were performed using a two-tailed, unpaired *t*-test. In experiments with more than two groups at a single time point, analysis was by one-way ANOVA with Tukey's multiple-comparison test. For data sets with multiple groups over a time course, analysis was performed using two-way ANOVA with either Tukey's or Bonferroni's multiple-comparison test as appropriate and indicated in the figure legends. Statistical analysis was performed using Prism 9.0 (GraphPad).

## Acknowledgements

We thank Tim Sparwasser for providing the *Tlr7*-floxed mice. We are indebted to the staff of the Imperial Central Biomedical Services for the care of the animals and the LMS/NIHR Imperial Biomedical Research Centre Flow Cytometry Facility for FACS support. CG was supported by an award from Imperial Institutional Strategic Support Fund. The research reported herein was supported by the Wellcome Trust (grant reference number: 108008/Z/15/Z [to MB] and 102126/B/13/Z [to CJ:AO]). CG was supported by an Imperial Wellcome Trust Institutional Strategic Support Fund and TCL by a Sir Henry Dale Fellowship from the Wellcome Trust and Royal Society (210424/Z/18/Z) and a KKLF project grant (KKL1379). We acknowledge contribution from the National Institute for Health Research (NIHR) Biomedical Research Centre based at Imperial College Healthcare NHS Trust and Imperial College London. The views expressed are those of the author(s) and not necessarily those of the NHS, the NIHR or the Department of Health.

## Additional information

### Funding

| Funder | Grant reference number | Author |
| --- | --- | --- |
| Wellcome Trust | 108008/Z/15/Z | Marina Botto |

| Funder | Grant reference number | Author |
|---|---|---|
| Wellcome Trust | 102126/B/13/Z | Cecilia Johansson |
| Wellcome Trust | Institutional Strategic Support Fund | Chiara Giacomassi |
| Wellcome Trust | Sir Henry Dale Fellowship - Wellcome Trust/Royal Society of Medicine 210424/Z/18/Z | Tiago C Luis |
| Kay Kendall Leukaemia Fund | KKL1379 | Tiago C Luis |
| Royal Society of Medicine | Sir Henry Dale Fellowship - Wellcome Trust/Royal Society of Medicine 210424/Z/18/Z | Tiago C Luis |

The funders had no role in study design, data collection and interpretation, or the decision to submit the work for publication. For the purpose of Open Access, the authors have applied a CC BY public copyright license to any Author Accepted Manuscript version arising from this submission.

## Author contributions

William D Jackson, Conceptualization, Data curation, Formal analysis, Investigation, Visualization, Methodology, Writing – original draft, Writing – review and editing; Chiara Giacomassi, Conceptualization, Data curation, Formal analysis, Investigation, Visualization, Methodology, Writing – review and editing; Sophie Ward, Amber Owen, Sarah Spear, Investigation; Tiago C Luis, Resources, Investigation, Methodology; Kevin J Woollard, Jessica Strid, Conceptualization, Supervision, Writing – review and editing; Cecilia Johansson, Resources, Supervision; Marina Botto, Conceptualization, Supervision, Funding acquisition, Writing – original draft, Writing – review and editing

## Author ORCIDs

Chiara Giacomassi http://orcid.org/0000-0003-0462-0381
Tiago C Luis http://orcid.org/0000-0002-6305-1257
Sarah Spear http://orcid.org/0000-0002-4306-3752
Kevin J Woollard http://orcid.org/0000-0002-9839-5463
Jessica Strid http://orcid.org/0000-0003-3690-2201
Marina Botto http://orcid.org/0000-0002-1458-3791

## Ethics

All procedures were carried out in accordance with the institutional guidelines and the studies were approved by the UK Home Office. Experimental studies were designed according the ARRIVE guidelines.

## Decision letter and Author response

Decision letter https://doi.org/10.7554/eLife.85647.sa1
Author response https://doi.org/10.7554/eLife.85647.sa2

## Additional files

### Supplementary files
• MDAR checklist

### Data availability
All data generated or analysed during this study are included in the manuscript and supporting file.

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
