## [Editor Report]

This work advances our understanding of TLR7 signalling at epithelial surfaces that drives monocytes expansion and its impact on viral responses. The evidence supporting this conclusion is solid, particularly data demonstrating TLR7 stimulation and the requirement for TLR7 in the monocyte expansion. The work will be of interest to immunologists and virologists.

---

## [Decision Letter]

**Decision letter after peer review:**

Thank you for submitting your article "TLR7 activation at epithelial barriers promotes emergency myelopoiesis and lung anti-viral immunity" for consideration by *eLife*. Your article has been reviewed by 2 peer reviewers, and the evaluation has been overseen by a Reviewing Editor and Paul Noble as the Senior Editor.

Essential revisions:

Both reviewers identified a few key aspects for revision. Please address them to the best of your ability.

*Reviewer #1 (Recommendations for the authors):*

Specific experiments that could increase the impact of this work and discussion points that could be clarified/reworded include:

1) Measurement of CSF-1 across the time course by protein in the serum/plasma of these mice, or in the bone marrow by RNA transcript, would rule in or out the most obvious candidate for the phenotype observed.

2) Inclusion of either further in vitro or in vivo experiments to demonstrate that the monocytes themselves are primed to respond differently to flu infection. Or rewording of the results/conclusion to better reflect the limitations of the experiments undertaken. For example, currently, the manuscript states: "Similarly, the extravasation of monocytes in the lung of R848-treated mice limited the viral replication after infection with respiratory syncytial virus (RSV) (Figure 7E), an RNA virus responsible for infant hospitalizations in the developed world". Unless I am totally misunderstanding the experiment this figure does not even show that monocytes, let alone extravasation of monocytes, is important in the changes in protection.

*Reviewer #2 (Recommendations for the authors):*

1. It is not clear from the figure legends if experiments were repeated or were a single study. This should be clarified in the manuscript.

2. Would recommend a more detailed phenotype of the respiratory viral pathogen lung responses. In particular, more detailed peripheral and lung immune phenotyping of monocytes and lung macrophages. This would better define the relationship of TLR7, myelopoiesis, and respiratory viral infection.

3. The images in Figure 6C are poor quality and hard to visualize. It is also not clear if they are intraparenchymal or intravascular.

---

## [Author Response]

Essential revisions:Reviewer #1 (Recommendations for the authors):Specific experiments that could increase the impact of this work and discussion points that could be clarified/reworded include:1) Measurement of CSF-1 across the time course by protein in the serum/plasma of these mice, or in the bone marrow by RNA transcript, would rule in or out the most obvious candidate for the phenotype observed.

We appreciate the point raised by the reviewer and CSF-1 was investigated (it was one the most obvious candidates for the monocytosis), but the results were not informative or inconclusive, and thus we decided not to include them in the paper. For the benefit of the reviewers we have listed below the additional experiments that we have conducted:

Trascriptomic analysis of the whole skin using nCounter Gene Expression System (NanoString technology): no increase in the Csf1 transcript counts were found in the R848-treated skin, making implausible for CSF-1 to be the key local factor driving the monocytosis (see Author response image 1). We did not consider measuring the Csf1 transcript in the bone marrow as we were looking for a skin-derived soluble factor connecting local TLR7 stimulation and the emergency myelopoiesis.Measurement of CSF-1 level in circulation. This analysis revealed increased serum levels of CSF-1 both after topical and intraperitoneal administration of R848 (see Author response image 1) making the involvement of CSF-1 unlikely as only activation of the TLR7 pathway at the skin barrier caused overt monocytosis, especially after 4 weeks of treatment.Blocking of CSF1R using the Ki20227 inhibitor: these experiments had to be stopped prematurely due to respiratory distress in the animals.Inhibition of the Glycoprotein 130 (gp130) protein that is part of the receptor signalling complexes for at least 8 cytokines (IL-6, IL-11, IL-27, LIF, CNTF, OSM, CT-1, and CLC) using the small-molecule SC144: these experiments had to be stopped after the second R848 treatment due to neurological toxicity of the compound (seizures).

**Author response image 1. sa2fig1:** (A) BALB/c mice (n=4 per group) received 2x topical ear treatments with 100ug R848, while vehicle mice received 2x treatments with acetone. Ear skin was harvested at 24 hours after the final treatment for transcriptomic analysis. Csf1 transcript counts shown are normalised to a panel of reference genes. Data represent a single experiment. (B) BALB/c mice (n=3 per group) received 100μg of R848 topically or I.P, 3x per week for 4 weeks. Control mice were given topical acetone. Serum CSF-1 levels were quantified by ELISA. Panel (A) was analysed by Mann-Whitney test; panel (B) by oneway ANOVA with Bonferroni’s multiple comparison test versus vehicle group. Data are the mean ± SEM; ns = non significant; p values are indicated.

2) Inclusion of either further in vitro or in vivo experiments to demonstrate that the monocytes themselves are primed to respond differently to flu infection. Or rewording of the results/conclusion to better reflect the limitations of the experiments undertaken. For example, currently, the manuscript states: "Similarly, the extravasation of monocytes in the lung of R848-treated mice limited the viral replication after infection with respiratory syncytial virus (RSV) (Figure 7E), an RNA virus responsible for infant hospitalizations in the developed world". Unless I am totally misunderstanding the experiment this figure does not even show that monocytes, let alone extravasation of monocytes, is important in the changes in protection.

The reviewer is correct and we apologise for the misleading sentence. We have rephrased this point in the main manuscript to reflect more accurately that our data show just a correlation and not a direct effect of extravasating monocytes on the local viral replication.

Reviewer #2 (Recommendations for the authors):1. It is not clear from the figure legends if experiments were repeated or were a single study. This should be clarified in the manuscript.

The number of repeated experiments is indicated at the end of each figure legend. When we have performed variable numbers, we have indicated the minimal number.

2. Would recommend a more detailed phenotype of the respiratory viral pathogen lung responses. In particular, more detailed peripheral and lung immune phenotyping of monocytes and lung macrophages. This would better define the relationship of TLR7, myelopoiesis, and respiratory viral infection.

We appreciate the point raised by the reviewer. In reply to the point # 2 of reviewer #1 we have amended the text to reflect more accurately the limitations of our experiments as they were just proof-of-concept. Further immune-phenotyping of lung monocytes and macrophages to assess their role in limiting the lung viral replication is beyond the scope of this manuscript. The monocytes that infiltrate the lungs may undergo a tissue adaptation and behave differently from the BM monocytes. Further studies will be required to define the specific adaptation of R848-treated monocytes following their migration in the lungs or other organs.

3. The images in Figure 6C are poor quality and hard to visualize. It is also not clear if they are intraparenchymal or intravascular.

We apologise for the reduced imagine quality of the PDF file. The original source image is of higher quality and has been uploaded in the *eLife* submission system. With regards to the macrophages being intravascular/intraparenchymal, this was addressed in figure 5D with the use of CD43 staining, where CD43^high^ cells are intravascular monocytes as described by Kuriakose et al., 2019. We have shown that CD43^high^ cells accumulate in the vasculature of the glomeruli, while CD68^high^ CD43^low^ remain in the kidney medulla.